# Mitigating Memorization of Noisy Labels via Regularization between Representations

**Hao Cheng**[*][†]**, Zhaowei Zhu**[*][†]**, Xing Sun**[§]**, and Yang Liu**[†]
[†]University of California, Santa Cruz, [§]Tencent YouTu Lab
{haocheng,zwzhu,yangliu}@ucsc.edu,
{winfred.sun}@gmail.com

## Abstract

Designing robust loss functions is popular in learning with noisy labels while existing designs did not explicitly consider the overfitting property of deep neural networks (DNNs). As a result, applying these losses may still suffer from overfitting/memorizing noisy labels as training proceeds. In this paper, we first theoretically analyze the memorization effect and show that a lower-capacity model may perform better on noisy datasets. However, it is non-trivial to design a neural network with the best capacity given an arbitrary task. To circumvent this dilemma, instead of changing the model architecture, we decouple DNNs into an encoder followed by a linear classifier and propose to restrict the function space of a DNN by a representation regularizer. Particularly, we require the distance between two self-supervised features to be positively related to the distance between the corresponding two supervised model outputs. Our proposed framework is easily extendable and can incorporate many other robust loss functions to further improve performance. Extensive experiments and theoretical analyses support our claims. Code is available at https://github.com/UCSC-REAL/SelfSup_NoisyLabel.

## 1 Introduction

Deep Neural Networks (DNNs) have achieved remarkable performance in many areas including speech recognition (Graves et al., 2013), computer vision (Krizhevsky et al., 2012; Lotter et al., 2016), natural language processing (Zhang & LeCun, 2015), *etc*. The high-achieving performance often builds on the availability of quality-annotated datasets. In a real-world scenario, data annotation inevitably brings in label noise (Wei et al., 2022d;e), which degrades the performance of the network, primarily due to DNNs' capability in "memorizing" noisy labels (Zhang et al., 2016).

In the past few years, a number of methods have been proposed to tackle the problem of learning with noisy labels. Notable achievements include robust loss design (Ghosh et al., 2017; Zhang & Sabuncu, 2018; Liu & Guo, 2020; Wang et al., 2021), sample selection (Han et al., 2018; Yu et al., 2019; Cheng et al., 2021; Xia et al., 2021b) , transition matrix estimation (Patrini et al., 2017; Zhu et al., 2021b; Xia et al., 2019; 2020b) and loss correction/reweighting based on noise transition matrix (Natarajan et al., 2013; Liu & Tao, 2015; Patrini et al., 2017; Jiang et al., 2021; Zhu et al., 2021b; Wei et al., 2022a; Zhu et al., 2022c). However, these methods still suffer from limitations because they are agnostic to the model complexity and do not explicitly take the over-fitting property of DNN into consideration when designing these methods (Wei et al., 2021; Liu et al., 2022). In the context of representation learning, DNN is prone to fit/memorize noisy labels as training proceeds (Wei et al., 2022d; Zhang et al., 2016), i.e., the memorization effect. Thus when the noise rate is high, even though the robust losses have some theoretical guarantees in expectation, they are still unstable during training (Cheng et al., 2021). It has been shown that early stopping helps mitigate memorizing noisy labels (Rolnick et al., 2017; Xia et al., 2020a). But intuitively, early stopping will handle overfitting wrong labels at the cost of underfitting clean samples if not tuned properly. An alternative approach is using regularizer to punish/avoid overfitting (Liu & Guo, 2020; Cheng et al.,

---

[*]Equal contributions.
[†]Corresponding author: Yang Liu yangliu@ucsc.edu.

2021; Liu et al., 2020), which mainly build regularizers by editing labels. In this paper, we study the effectiveness of a representation regularizer.

To fully understand the memorization effect on learning with noisy labels, we decouple the generalization error into estimation error and approximation error. By analyzing these two errors, we find that DNN behaves differently on various label noise types and the key to prevent over-fitting is to control model complexity. However, specifically designing the model structure for learning with noisy labels is hard. One tractable solution is to use representation regularizers to cut off some redundant function space without hurting the optima. Therefore, we propose a unified framework by utilizing representation to mitigate the memorization effect. We list main contributions below:

- We first theoretically analyze the memorization effect by decomposing the generalization error into estimation error and approximation error in the context of learning with noisy labels and show that a lower-capacity model may perform better on noisy datasets.
- Due to the fact that designing a neural network with the best capacity given an arbitrary task requires formidable effort, instead of changing the model architecture, we decouple DNNs into an encoder followed by a linear classifier and propose to restrict the function space of DNNs by the structural information between representations. Particularly, we require the distance between two self-supervised features to be positively related to the distance between the corresponding two supervised model outputs.
- The effectiveness of the proposed regularizer is demonstrated by both theoretical analyses and numerical experiments. Our framework can incorporate many current robust losses and help them further improve performance.

## 1.1 RELATED WORKS

**Learning with Noisy Labels** Many works design robust loss to improve the robustness of neural networks when learning with noisy labels (Ghosh et al., 2017; Zhang & Sabuncu, 2018; Liu & Guo, 2020; Xu et al., 2019; Feng et al., 2021; Yong et al.; Xia et al., 2022; Wei et al., 2022c;b). (Ghosh et al., 2017) proves MAE is inherently robust to label noise. However, MAE has a severe under-fitting problem. (Zhang & Sabuncu, 2018) proposes GCE loss which can combine the advantage of MAE and CE, exhibiting good performance on noisy datasets. (Liu & Guo, 2020) introduces peer loss, which is statistically robust to label noise without knowing noise rates. The extension of peer loss also shows good performance on instance-dependent label noise (Cheng et al., 2021; Zhu et al., 2021a). Another efficient approach to combat label noise is by sample selection (Jiang et al., 2018; Han et al., 2018; Yu et al., 2019; Northcutt et al., 2021; Yao et al., 2020; Wei et al., 2020; Zhang et al., 2020; Xia et al., 2021a). These methods regard "small loss" examples as clean ones and train multiple networks to select clean samples. Semi-supervised learning is also popular and effective on learning with noisy labels in recent years. Some works (Li et al., 2020; Nguyen et al., 2020) perform clustering on the sample loss and divide the samples into clean ones and noisy ones. Then drop the labels of the "noisy samples" and perform semi-supervised learning on all the samples. However, the semi-supervised pseudo labels can cause disparate impact on different groups of data Zhu et al. (2022b). Recently, some works apply self-supervised learning to handle noisy labels (Ghosh & Lan, 2021; Li et al., 2022a; Wei et al., 2023). Our work can also explain some findings from (Ghosh & Lan, 2021).

**Knowledge Distillation** Our proposed learning framework is related to knowledge distillation (KD). (Hinton et al., 2015) shows that a small, shallow network can be improved through a teacher-student framework. Due to its great applicability, KD has gained more and more attention in recent years and numerous methods have been proposed to perform efficient distillation (Mirzadeh et al., 2020; Zhang et al., 2018b; 2019). However, the dataset used in KD is assumed to be clean. Thus it is hard to connect KD with learning with noisy labels. In this paper, we theoretically and experimentally show that a regularizer generally used in KD (Park et al., 2019) can alleviate the over-fitting problem on noisy data by using DNN features which offers a new alternative for dealing with label noise.

## 2 PRELIMINARY

We introduce preliminaries and notations including definitions and problem formulation.

**Problem formulation** Consider a $K$-class classification problem on a set of $N$ training examples denoted by $D := \{(x_n, y_n)\}_{n \in [N]}$, where $[N] := \{1, 2, \cdots, N\}$ is the set of example indices.

Examples $(x_n, y_n)$ are drawn according to random variables $(X, Y)$ from a joint distribution $\mathcal{D}$. The classification task aims to identify a classifier $C$ that maps $X$ to $Y$ accurately. In real-world applications, the learner can only observe noisy labels $\tilde{y}$ drawn from $\widetilde{Y}|X$ (Wei et al., 2022d), e.g., human annotators may wrongly label some images containing cats as ones that contain dogs accidentally or irresponsibly. The corresponding noisy dataset and distribution are denoted by $\widetilde{D} := \{(x_n, \tilde{y}_n)\}_{n \in [N]}$ and $\widetilde{\mathcal{D}}$. Define the expected risk of a classifier $C$ as $R(C) = \mathbb{E}_{\mathcal{D}}\left[\mathbb{1}(C(X) \neq Y)\right]$. The goal is to learn a classifier $C$ from the noisy distribution $\widetilde{\mathcal{D}}$ which also minimizes $R(C)$, i.e., learn the *Bayes optimal classifier* such that $C^*(x) = \arg\max_{i \in [K]} \mathbb{P}(Y = i | X = x)$.

**Noise transition matrix**  The label noise of each instance is characterized by $T_{ij}(X) = \mathbb{P}(\widetilde{Y} = j | X, Y = i)$, where $T(X)$ is called the (instance-dependent) noise transition matrix (Zhu et al., 2021b; Li et al., 2022b). There are two special noise regimes (Han et al., 2018) for the simplicity of theoretical analyses: *symmetric noise* and *asymmetric noise*. In symmetric noise, each clean label is randomly flipped to the other labels uniformly w.p. $\epsilon$, where $\epsilon$ is the noise rate. Therefore, $T_{ii} = 1 - \epsilon$ and $T_{ij} = \frac{\epsilon}{K-1}, i \neq j, i, j \in [K]$. In asymmetric noise, each clean label is randomly flipped to its adjacent label, i.e., $T_{ii} = 1 - \epsilon, T_{ii} + T_{i,(i+1)_K} = 1$, where $(i+1)_K := i \mod K + 1$.

**Empirical risk minimization**  The empirical risk on a noisy dataset with classifier $C$ writes as $\frac{1}{N}\sum_{n \in [N]} \ell(C(x_n), \tilde{y}_n)$, where $\ell$ is usually the cross-entropy (CE) loss. Existing works adapt $\ell$ to make it robust to label noise, e.g., loss correction (Natarajan et al., 2013; Patrini et al., 2017), loss reweighting (Liu & Tao, 2015), generalized cross-entropy (GCE) (Zhang & Sabuncu, 2018), peer loss (Liu & Guo, 2020), $f$-divergence (Wei & Liu, 2021). To distinguish their optimization from the vanilla empirical risk minimization (ERM), we call them the adapted ERM.

**Memorization effects of DNNs**  Without special treatments, minimizing the empirical risk on noisy distributions make the model overfit the noisy labels. As a result, the corrupted labels will be memorized (Wei et al., 2022d; Han et al., 2020; Xia et al., 2020a) and the test accuracy on clean data will drop in the late stage of training even though the training accuracy is consistently increasing. See Figure 1 for an illustration. Therefore, it is important to study robust methods to mitigate memorizing noisy labels.

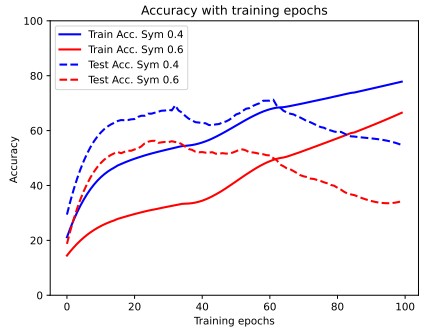

**Outline**  The rest of the paper is organized as follows. In Section 3, we theoretically understand the memorization effect by analyzing the relationship among noise rates, sample size, and model capacity, which motivate us to design a regularizer to alleviate the memorization effect in Section 4 by restricting model capacity. Section 5 empirically validates our analyses and proposal.

Figure 1: Training and test accuracies on CIFAR-10 with symmetric noise with noise rates 0.4 (blue curves) and 0.6 (red curves). We use ResNet 34 for conducting the experiments (See detailed setting in Appendix E)

## 3   IMPACTS OF MISLABELED DATA ON DNN PERFORMANCE

We quantify the harmfulness of memorizing noisy labels by analyzing the generalization errors on clean data when learning on a noisy dataset $\widetilde{D}$ and optimizing over function space $\mathcal{C}$.

### 3.1   THEORETICAL TOOLS

Denote by the optimal clean classifier $C_{\mathcal{D}} := \arg\min_{C \in \mathcal{C}} \mathbb{E}_{\mathcal{D}}[\ell(C(X), Y)]$, the optimal noisy classifier $C_{\widetilde{\mathcal{D}}} = \arg\min_{C \in \mathcal{C}} \mathbb{E}_{\widetilde{\mathcal{D}}}[\ell(C(X), \widetilde{Y})]$, and the learned classifier on the noisy dataset $\widehat{C}_{\widetilde{D}} = \arg\min_{C \in \mathcal{C}} \sum_{n \in [N]}[\ell(C(x_n), \tilde{y}_n)]$. The expected risk w.r.t the Bayes optimal classifier $C^*$ can be decomposed into two parts: $\mathbb{E}[\ell(\widehat{C}_{\widetilde{D}}(f(X)), Y)] - \mathbb{E}[\ell(C^*(X), Y)] = \mathsf{Error}_E(C_{\mathcal{D}}, \widehat{C}_{\widetilde{D}}) + \mathsf{Error}_A(C_{\mathcal{D}}, C^*)$, where the estimation error $\mathsf{Error}_E$ and the approximation error $\mathsf{Error}_A$ can be written as $\mathsf{Error}_E(C_{\mathcal{D}}, \widehat{C}_{\widetilde{D}}) = \mathbb{E}[\ell(\widehat{C}_{\widetilde{D}}(X), Y)] - \mathbb{E}[\ell(C_{\mathcal{D}}(X), Y)]$, $\mathsf{Error}_A(C_{\mathcal{D}}, C^*) = \mathbb{E}[\ell(C_{\mathcal{D}}(X), Y)] - \mathbb{E}[\ell(C^*(X), Y)]$. We analyze each part respectively.

**Estimation error**  We first study the noise consistency from the aspect of expected loss.

**Definition 1** (Noise consistency). *One label noise regime satisfies the noise consistency under loss $\ell$ if the following affine relationship holds: $\mathbb{E}_{\widetilde{\mathcal{D}}}[\ell(C(X), \widetilde{Y})] = \gamma_1 \mathbb{E}_{\mathcal{D}}[\ell(C(X), Y)] + \gamma_2$, where $\gamma_1 > 0$ and $\gamma_2$ are constants in a fixed noise setting.*

Following the probabilistic decomposition of label noise Natarajan et al. (2013); Ghosh et al. (2017); Cheng et al. (2021), we have Lemmas 1 and 2.

**Lemma 1.** *A general noise regime with noise transitions $T_{ij}(X) : \mathbb{P}(\widetilde{Y} = j | Y = i, X)$ can be decoupled to the following form:*

$$\mathbb{E}_{\widetilde{\mathcal{D}}}\left[\ell(C(X), \widetilde{Y})\right] = \underline{T} \cdot \mathbb{E}_{\mathcal{D}}[\ell(C(X), Y)] + \sum_{j \in [K]} \sum_{i \in [K]} \mathbb{P}(Y = i) \mathbb{E}_{\mathcal{D}|Y=i}[U_{ij}(X)\ell(C(X), j)],$$

*where $U_{ij}(X) = T_{ij}(X), \forall i \neq j, U_{jj}(X) = T_{jj}(X) - \underline{T}, \underline{T} := \min_{X,i} T_{ii}(X).$*

Lemma 1 shows the general instance-dependent label noise is hard to be consistent since the second term is not a constant unless we add more restrictions to $T(X)$. Specially, in Lemma 2, we consider two typical noise regimes for multi-class classifications: symmetric noise and asymmetric noise.

**Lemma 2.** *The symmetric noise is consistent with 0-1 loss: $\mathbb{E}_{\widetilde{\mathcal{D}}}\left[\ell(C(X), \widetilde{Y})\right] = \gamma_1 \mathbb{E}_{\mathcal{D}}[\ell(C(X), Y)] + \gamma_2$, where $\gamma_1 = \left(1 - \frac{\epsilon K}{K-1}\right)$, $\gamma_2 = \frac{\epsilon}{K-1}$. The asymmetric noise is not consistent: $\mathbb{E}_{\widetilde{\mathcal{D}}}\left[\ell(C(X), \widetilde{Y})\right] = (1 - \epsilon) \cdot \mathbb{E}_{\mathcal{D}}[\ell(C(X), Y)] + \epsilon \sum_{i \in [K]} \mathbb{P}(Y = i) \mathbb{E}_{\mathcal{D}|Y=i}[\ell(C(X), (i+1)_K)].$*

With Lemma 2, we can upper bound the estimation errors in Theorem 1.

**Theorem 1.** *With probability at least $1 - \delta$, learning with symmetric/asymmetric noise and 0-1 loss has the following estimation error:*

$$\textit{Error}_E(C_{\mathcal{D}}, \widehat{C}_{\widetilde{D}}) \leq \Delta_E(\mathcal{C}, \varepsilon, \delta) := 16\sqrt{\frac{|\mathcal{C}| \log(N \cdot e/|\mathcal{C}|) + \log(8/\delta)}{2N(1 - \varepsilon)^2}} + \textit{Bias}(C_{\mathcal{D}}, \widehat{C}_{\widetilde{D}}),$$

*where $e = 2.718$ is the base of the natural logarithms, $|\mathcal{C}|$ is the VC-dimension of function class $\mathcal{C}$ (Bousquet et al., 2003; Devroye et al., 2013). The noise rate parameter $\varepsilon$ satisfies $\varepsilon = \frac{\epsilon K}{K-1}$ for symmetric noise and $\varepsilon = \epsilon$ for asymmetric noise. The bias satisfies $\textit{Bias}(C_{\mathcal{D}}, \widehat{C}_{\widetilde{D}}) = 0$ for symmetric noise and $\textit{Bias}(C_{\mathcal{D}}, \widehat{C}_{\widetilde{D}}) = \frac{\epsilon}{1-\epsilon} \sum_{i \in [K]} \mathbb{P}(Y = i) \mathbb{E}_{\mathcal{D}|Y=i}[\ell(C_{\mathcal{D}}(X), (i+1)_K) - \ell(\widehat{C}_{\widetilde{D}}(X), (i+1)_K)]$ for asymmetric noise.*

**Approximation error** Analyzing the approximation error for an arbitrary DNN is an open-problem and beyond our scope. Generally, according to the trade-off between approximation error and estimation error (a.k.a. bias-complexity trade-off (Shalev-Shwartz & Ben-David, 2014)), a large function space $\mathcal{C}$ reduces the approximation error at the cost of increasing the estimation error. Note an alternative of the generalization bound is using the Rademacher complexity of $\mathcal{C}$. Both bounds help reveal the tradeoff between approximation error and estimation error w.r.t $\mathcal{C}$. We use VC-dimension since it shows a clearer relationship between model capacity $\mathcal{C}$ (numerator) and the effective number of samples $N(1 - \varepsilon)^2$ (denominator).

**Trade-off** From Theorem 1 and the above analyses, the bias complexity trade-off is more severe in the presence of label noise. In the case of symmetric label noise, a larger $|\mathcal{C}|$ will lead to *smaller* approximation error but at the cost of *larger* estimation error given large $N$. Although the trade-off may not be remarkable in traditional learning with clean data ($\epsilon = 0$), it is critical when label noise exists since the existence of $\epsilon$ will significantly reduce the effective datasize from $N$ to $N(1 - \epsilon)^2$. In practice, it is non-trivial to find the best function space or design the best neural network given an arbitrary task for reducing the total generalization error. We will introduce a tractable solution in the following sections.

## 3.2 DECOUPLED CLASSIFIERS: FROM FUNCTION SPACES TO REPRESENTATIONS

One tractable way to restrict the function space is fixing some layers of a given DNN model. Particularly, we can decouple $C$ into two parts: $C = f \circ g$, where the encoder $f$ extract representations from

Figure 2: Illustration of different learning paths (distinguished by colors). The curve with arrow between two green dots indicates the effort (e.g., number of training instances) of training a model from one state to another state.

raw features and the linear classifier $g$ maps representations to label classes, i.e., $C(X) = g(f(X))$. Clearly, the function space can be reduced significantly if we only optimize the linear classifier $g$. But the performance of the classifier depends heavily on the encoder $f$. By this decomposition, we transform the problem of finding good *function spaces* to finding good *representations*.

Now we analyze the effectiveness of such decomposition. Figure 2 illustrates three learning paths. *Path-1* is the traditional learning path that learns both encoder $f$ and linear classifier $g$ at the same time (Patrini et al., 2017). In *Path-2*, a pre-trained encoder $f$ is adopted as an initialization of DNNs and both $f$ and $g$ are fine-tuned on noisy data distributions $\widetilde{\mathcal{D}}$ (Ghosh & Lan, 2021). The pre-trained encoder $f$ is also adopted in *Path-3*. But the encoder $f$ is fixed/frozen throughout the later training procedures and only the linear classifier $g$ is updated with $\widetilde{\mathcal{D}}$. We compare the generalization errors of different paths to provide insights for the effects of representations on learning with noisy labels.

Now we instantiate function spaces $\mathcal{C}_1$ and $\mathcal{C}_2$ with different representations. With traditional training or an unfixed encoder (Path-1 or Path-2), classifier $C$ is optimized over function space $\mathcal{C}_1 = \mathcal{G} \circ \mathcal{F}$ with raw data. With a fixed encoder (Path-3), classifier $C$ is optimized over function space $\mathcal{G}$ given representations $f(X)$.

**Symmetric noise** Let $\mathcal{C}_1 = \mathcal{G} \circ \mathcal{F}$, $\mathcal{C}_2 = \mathcal{G}|f$. Denote the optimal classifier learned within the above two functions spaces by $C_{\mathcal{D}}^{\mathcal{G} \circ \mathcal{F}}$ and $C_{\mathcal{D}}^{\mathcal{G}|f}$, respectively. Then the approximation errors of both cases can be denoted by $\mathsf{Error}_A(C_{\mathcal{D}}^{\mathcal{G} \circ \mathcal{F}}, C^*)$ and $\mathsf{Error}_A(C_{\mathcal{D}}^{\mathcal{G}|f}, C^*)$. Assume $\mathsf{Error}_A(C_{\mathcal{D}}^{\mathcal{G} \circ \mathcal{F}}, C^*) < \mathsf{Error}_A(C_{\mathcal{D}}^{\mathcal{G}|f}, C^*)$. Note the assumption holds generally for analyzing the bias-complexity trade-off; otherwise we should always prefer a fixed encoder.

With the error proxy $\Delta_E(\mathcal{C}, \varepsilon, \delta)$ in Theorem 1, we reveal the relationship among total generalization error, model capacity and noise rate in Corollary 1.

**Corollary 1.** *When $\mathsf{Error}_A(C_{\mathcal{D}}^{\mathcal{G} \circ \mathcal{F}}, C^*) < \mathsf{Error}_A(C_{\mathcal{D}}^{\mathcal{G}|f}, C^*)$, the error proxy of the expected generalization error for the fixed encoder ($\mathcal{G}|f$) is not greater than that for the unfixed one ($\mathcal{G} \circ \mathcal{F}$) when*

$$1 - \frac{\epsilon K}{K-1} \le \beta'(\mathcal{G} \circ \mathcal{F}, \mathcal{G}|f) := \frac{16}{\sqrt{2N}} \frac{\left( \sqrt{|\mathcal{G} \circ \mathcal{F}| \log(4N \cdot e/|\mathcal{G} \circ \mathcal{F}|)} - \sqrt{|\mathcal{G}| \log(4N \cdot e/|\mathcal{G}|)} \right)}{\mathsf{Error}_A(C_{\mathcal{D}}^{\mathcal{G}|f}, C^*) - \mathsf{Error}_A(C_{\mathcal{D}}^{\mathcal{G} \circ \mathcal{F}}, C^*)}.$$

Recall in above $|\cdot|$ for a hypothesis space denotes its VC-dimension. RHS is greater than 0 and LHS is decreasing with the increase of $\epsilon$. Corollary 1 implies that, for symmetric noise, a fixed encoder is likely better in high-noise settings. Section 5 provides empirical results to validate this claim. we also provide empirical evidence in the Appendix D.4 that a shallow network (low capacity model, similar to fixing the encoder) performs better than deeper network for high-noise regimes.

**Other noise** Based on Theorem 1, for asymmetric label noise, the noise consistency is broken and the bias term makes the learning error hard to be bounded. As a result, the relationship between noise rate $\epsilon$ and generalization error is not clear and simply fixing the encoder may induce a larger generalization error. For the general instance-dependent label noise, the bias term is more complicated thus the benefit of fixing the encoder is less clear.

**Insights and Takeaways** With the above analyses, we know learning with an unfixed encoder is not stable, which is easier to be affected by noisy patterns and yields a worse result than a properly selected fixed encoder when the noise rate is high. Restricting the search space makes the convergence stable (reducing estimation error) with the cost of increasing approximation errors. This motivates us to find a way to compromise between a fixed and unfixed encoder. We explore towards this direction in the next section.

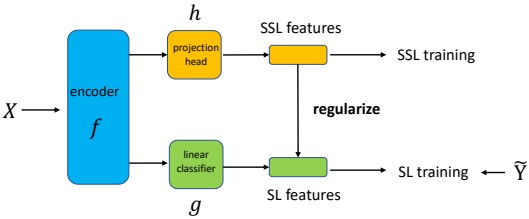

Figure 3: The training framework of using representations (SSL features) to regularize learning with noisy labels (SL features).

## 4 COMBATING MEMORIZATION BY REPRESENTATION REGULARIZATION

Our understandings in Section 3 motivate us to use the information from representations to regularize the model predictions. Intuitively, as long as the encoder is not fixed, the approximation error could be low enough. If the ERM is properly regularized, the search space and the corresponding estimation error could be reduced.

### 4.1 TRAINING FRAMEWORK

The training framework is shown in Figure 3, where a new learning path (Self-supervised learning, SSL) $f \to h$ is added to be parallel to Path-2 $f \to g$ (SL-training) in Figure 2. The newly added *projection head* $h$ is one-hidden-layer MLP (Multi Layer Perceptron) whose output represents SSL features (after dimension reduction). Its output is employed to regularize the output of linear classifier $g$. Given an example $(x_n, \tilde{y}_n)$ and a random batch of features $\mathcal{B}$ ($x_n \in \mathcal{B}$), the loss is defined as:

$$L((x_n, \tilde{y}_n); f, g, h) = \underbrace{\ell(g(f(x_n)), \tilde{y}_n)}_{\text{SL Training}} + \underbrace{\ell_{\text{Info}}(h(f(x_n)), \mathcal{B})}_{\text{SSL Training}} + \lambda \underbrace{\ell_{\text{Reg}}(h(f(x_n)), g(f(x_n)), \mathcal{B})}_{\text{Representation Regularizer}}, \quad (1)$$

where $\lambda$ controls the scale of regularizer. The loss $\ell$ for SL training could be either the traditional CE loss or recent robust loss such as loss correction/reweighting (Patrini et al., 2017; Liu & Tao, 2015), GCE (Zhang & Sabuncu, 2018), peer loss (Liu & Guo, 2020). The SSL features are learned by InfoNCE (Van den Oord et al., 2018):

$$\ell_{\text{Info}}(h(f(x_n)), \mathcal{B}) := -\log \frac{\exp(\text{sim}(h(f(x_n)), h(f(x'_n))))}{\sum_{x_{n'} \in \mathcal{B}, n' \neq n} \exp(\text{sim}(h(f(x_n)), h(f(x_{n'}))))}.$$

Note InfoNCE and CE share a common encoder, inspired by the design of self distillation (Zhang et al., 2019). The regularization loss $\ell_{\text{Reg}}$ writes as:

$$\ell_{\text{Reg}}(h(f(x_n)), g(f(x_n)), \mathcal{B}) = \frac{1}{|\mathcal{B}| - 1} \sum_{x_{n'} \in \mathcal{B}, n \neq n'} d(\phi^w(t_n, t_{n'}), \phi^w(s_n, s_{n'})),$$

where $d(\cdot)$ is a distance measure for two inputs, e.g., $l_1$ or square $l_2$ distance, $t_n = h(f(x_n))$, $s_n = g(f(x_n))$, $\phi^w(t_n, t_{n'}) = \frac{1}{m}\|t_n - t_{n'}\|^w$, where $w \in \{1, 2\}$ represents $l_1$ norm and squared $l_2$ metric, $m$ normalizes the distance over a batch:

$$m = \frac{1}{|\mathcal{B}|(|\mathcal{B}| - 1)} \sum_{x_n, x_{n'} \in \mathcal{B}, n \neq n'} \|t_n - t_{n'}\|^w. \quad (2)$$

The design of $\ell_{\text{Reg}}$ follows the idea of clusterability (Zhu et al., 2021b; 2022a) and inspired by relational knowledge distillation (Park et al., 2019), i.e., **instances with similar SSL features should have the same true label and instance with different SSL features should have different true labels**, which is our motivation to design $\ell_{\text{Reg}}$. Due to the fact that SSL features are learned from raw feature $X$ and independent of noisy label $\tilde{Y}$, then using SSL features to regularize SL features is supposed to mitigate memorizing noisy labels. We provide more theoretical understandings in the following subsection to show the effectiveness of this design.

### 4.2 THEORETICAL UNDERSTANDING

We theoretically analyze how $\ell_{\text{reg}}$ mitigates memorizing noisy labels in this subsection. As we discussed previously, SSL features are supposed to pull the model away from memorizing wrong labels due to clusterability (Zhu et al., 2021b). However, since the SL training is performed on the

noisy data, when it achieves zero loss, the minimizer should be either memorizing each instance (for CE loss) or their claimed optimum (for other robust loss functions). Therefore, the global optimum should be at least affected by both SL training and representation regularization, where the scale is controlled by $\lambda$. For a clear presentation, we focus on analyzing the effect of $\ell_{\text{reg}}$ in a binary classification, whose minimizer is approximate to the global minimizer when $\lambda$ is sufficiently large.

Consider a randomly sampled batch $\mathcal{B}$. Denote by $\mathcal{X}^2 := \{(x_i, x_j)|x_i \in \mathcal{B}, x_j \in \mathcal{B}, i \neq j\}$ the set of data pairs, and $d_{i,j} = d(\phi^w(t_i, t_j), \phi^w(s_i, s_j))$. The regularization loss of batch $\mathcal{B}$ is decomposed as:

$$\frac{1}{|\mathcal{B}|} \sum_{n|x_n \in \mathcal{B}} \ell_{\text{Reg}}(h(f(x_n)), g(f(x_n)), \mathcal{B}) = \frac{1}{|\mathcal{X}^2|} \Big( \underbrace{\sum_{(x_i, x_j) \in \mathcal{X}_{\text{T}}^2} d_{i,j}}_{\text{Term-1}} + \underbrace{\sum_{(x_i, x_j) \in \mathcal{X}_{\text{F}}^2} d_{i,j}}_{\text{Term-2}} + \underbrace{\sum_{x_i \in \mathcal{X}_{\text{T}}, x_j \in \mathcal{X}_{\text{F}}} 2d_{i,j}}_{\text{Term-3}} \Big). \quad (3)$$

where $\mathcal{X} = \mathcal{X}_{\text{T}} \bigcup \mathcal{X}_{\text{F}}$, $\mathcal{X}_{\text{T}}/\mathcal{X}_{\text{F}}$ denotes the set of instances whose labels are true/false. Note the regularizer mainly works when SSL features "disagree" with SL features, i.e., Term-3. Denote by

$$X_+ \sim \mathbb{P}(X|Y=1), \quad X_- \sim \mathbb{P}(X|Y=0), \quad X^{\text{T}} \sim \mathbb{P}(X|Y=\widetilde{Y}), \quad X^{\text{F}} \sim \mathbb{P}(X|Y \neq \widetilde{Y}).$$

For further analyses, we write Term-3 in the form of expectation with $d$ chosen as square $l_2$ distance, *i.e.*, MSE loss:

$$L_c = \mathbb{E}_{X^{\text{T}}, X^{\text{F}}} \left( \frac{||g(f(X^{\text{T}})) - g(f(X^{\text{F}}))||^1}{m_1} - \frac{||h(f(X^{\text{T}})) - h(f(X^{\text{F}}))||^2}{m_2} \right)^2, \quad (4)$$

where $m_1$ and $m_2$ are normalization terms in Eqn (2). Note in $L_c$, we use $w = 1$ for SL features and $w = 2$ for SSL features.[1] Denote the variance by $\text{var}(\cdot)$. In the setting of binary classification, define notations: $X_+^{\text{F}} \sim \mathbb{P}(X|\widetilde{Y}=1, Y=0)$, $X_-^{\text{F}} \sim \mathbb{P}(X|\widetilde{Y}=0, Y=1)$.

To find a tractable way to analytically measure and quantify how feature correction relates to network robustness, we make three assumptions as follows:

**Assumption 1** (Memorize clean instances). $\forall n \in \{n|\tilde{y}_n = y_n\}, \ell(g(f(x_n)), y_n) = 0$.

**Assumption 2** (Same overfitting). $\text{var}(g(f(X_+^F))) = 0$ *and* $\text{var}(g(f(X_-^F))) = 0$.

**Assumption 3** (Gaussian-distributed SSL features). *The SSL features follow Gaussian distributions, i.e.*, $h(f(X_+)) \sim \mathcal{N}(\mu_1, \Sigma)$ *and* $h(f(X_-)) \sim \mathcal{N}(\mu_2, \Sigma)$, *where $\Sigma$ is the covariance matrix.*

Assumption 1 implies that a DNN has confident predictions on clean samples. Assumption 2 implies that a DNN has the same degree of overfitting for different classes of noisy samples. For example, an over-parameterized DNN can memorize all the noisy labels (Zhang et al., 2016), which is the focus of this paper. Thus these two assumptions are reasonable (we also provide empirical evidence for Assumption 2 in Appendix D.5. Assumption 3 assumes that SSL features follow Gaussian distribution when we add the regularize. Intuitively, to provide useful regularization, the SSL features should not be arbitrarily bad. In our experiments, we find that SSL features of CIFAR10 are relatively good, thus regularizer can be added in the very beginning while for CIFAR100, we need certain warmup epochs before adding the regularizer. Next, we present Theorem 2 to analyze the effect of $L_c$. Consider the case that the model is over-parameterized and traditional training can memorize all samples. Let $e_+ = \mathbb{P}(\widetilde{Y}=0|Y=1), e_- = \mathbb{P}(\widetilde{Y}=1|Y=0)$, we have:

**Theorem 2.** *Based on Assumptions 1–3, when $e_- = e_+ = e$, $\mathbb{P}(Y=1) = \mathbb{P}(Y=0)$, assuming that the Bayes classifier achieves zero error, $N$ and $\mathcal{B}$ are sufficiently large, then minimizing $L_c$ over $g, h$, and $f$ on DNN results in:*

$$\mathbb{E}_{\mathcal{D}} \left[ \mathbb{1} \left( g^*(f^*(X), Y) \right] = e \cdot \left( \frac{1}{2} - \frac{1}{2 + \Delta(\Sigma, \mu_1, \mu_2)} \right) \right.$$

*where $\Delta(\Sigma, \mu_1, \mu_2) := 8 \cdot tr(\Sigma)/||\mu_1 - \mu_2||^2$, $tr(\cdot)$ denotes the matrix trace, and $g^*, f^*$ denote the optimal model.*

Theorem 2 reveals a clean relationship between the quality of SSL features (given by $h(f(X))$) and the network robustness on noisy samples. When $tr(\Sigma) \to 0$ or $||\mu_1 - \mu_2|| \to \infty$, the expected risk of the model $\mathbb{E}_{\mathcal{D}} [\mathbb{1} (g(f(X), Y)]$ will approximate to 0. *I.e.,* for any sample $x$, the model will predict $x$ to its clean label. Note the proof of Theorem 2 does not rely on any SSL training process. This makes it possible to use some pre-trained encoders from other tasks. In the Appendix, we also provide an theoretical understanding on the regularizer from the perspective of information theory.

---

[1]Practically, different choices make negligible effects on performance. See more details in Appendix.

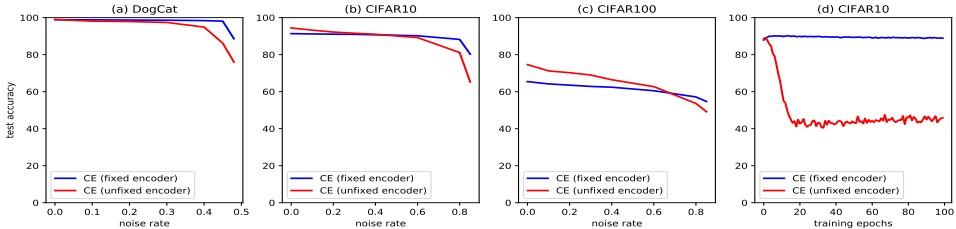

Figure 5: (a) (b) (c): Performance of CE on DogCat, CIFAR10 and CIFAR100 under symmetric noise rate. For each noise rate, the best epoch test accuracy is recorded. The blue line represents training with fixed encoder and the red line represents training with unfixed encoder; (d): test accuracy of CIFAR10 on each training epoch under symmetric 0.6 noise rate. We use ResNet50 (He et al., 2016) for DogCat and ResNet34 for CIFAR10 and CIFAR100. Encoder is pre-trained by SimCLR (Chen et al., 2020). Detailed settings are reported in the Appendix.

## 5 EMPIRICAL EVIDENCES

### 5.1 THE EFFECT OF REPRESENTATIONS

We perform experiments to study the effect of representations on learning with noisy labels. Figure 5 shows the learning dynamics on symmetric label noise while Figure 4 shows the learning dynamics on asymmetric and instance-dependent label noise. From these two figures, given a good representation, we have some key observations:

- **Observation-1:** Fix encoders for high symmetric label noise
- **Observation-2:** Do not fix encoders for low symmetric label noise
- **Observation-3:** Do not fix encoder when bias exists
- **Observation-4:** A fixed encoder is more stable during learning

**Observation-1**, **Observation-2** are verified by Figure 5 (a) (b) (c) and **Observation-3** is verified by Figure 4(a) (b). **Observation-4** is verified by Figure 5 (d). These four observations are consistent with our analyses in Section 3. We also find an interesting phenomenon in Figure 4 (b) that down-sampling (making $\mathbb{P}(\widetilde{Y} = i) = \mathbb{P}(\widetilde{Y} = j)$ in the noisy dataset) is very helpful for instance-dependent label noise since down-sampling can reduce noise rate imbalance (we provide an illustration on binary case in the Appendix) which could lower down the estimation error. Ideally, if down-sampling could make noise-rate pattern be symmetric, we could achieve noise consistency (Definition 1) which results in 0 Bias from Theorem 1. Another interesting phenomenon

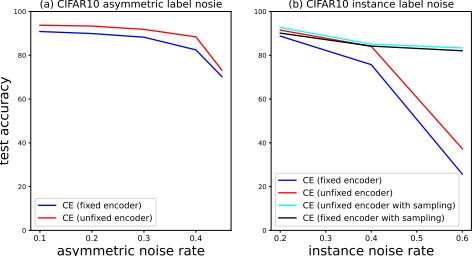

Figure 4: (a) performance of CE on asymmetric label noise. (b) performance of CE on instance-dependent label noise. The generation of instance-dependent label noise follows from CORES (Cheng et al., 2021).

is that from Figure 5 (a) (b) (c), the crossing point is different for each dataset. This phenomenon can be explained by Corollary 1. Corollary 1 implies that if the encoder is learned very well, i.e., $\mathsf{Error}_A(C_{\mathcal{D}}^{\mathcal{G} \circ \mathcal{F}}, C^*) \approx \mathsf{Error}_A(C_{\mathcal{D}}^{\mathcal{G}|f}, C^*)$, fixing the encoder has benefits over unfixed encoder even when noise rate is small. Since for DogCat, CIFAR10 and CIFAR100 dataset, each class have 12500 samples, 5000 samples and 500 samples, respectively. When applying the self-supervised learning on these datasets, the encoder quality is DogCat > CIFAR10 > CIFAR100. Thus the crossing point is small for DogCat and large for CIFAR100.

### 5.2 THE PERFORMANCE OF USING REPRESENTATIONS AS A REGULARIZER

**Experiments on synthetic label noise** We first show that Regularizer can alleviate the over-fitting problem when $\ell$ in Equation (1) is simply chosen as Cross-Entropy loss. The experiments are shown in Figure 6. Regularizer is added at the very beginning since recent studies show that for a randomly initialized network, the model tends to fit clean labels first (Arpit et al., 2017) and we hope the regularizer can improve the network robustness when DNN begins to fit noisy labels. From Figure 6 (c) (d), for CE training, the performance first increases then decreases since the network over-fits noisy labels as training proceeds. However, for CE with the regularizer, the performance is more stable after it reaches the peak. For 60% noise rate, the peak point is also much higher than vanilla CE training. For Figure 6 (a) (b), since the network is not randomly initialized, it over-fits noisy labels at

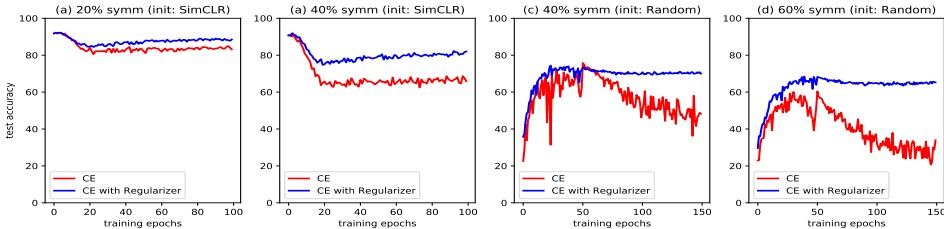

Figure 6: Experiments *w.r.t.* regularizer ($\lambda = 1$) on CIFAR10. ResNet34 is deployed for the experiments. (a) (b): Encoder is pre-trained by SimCLR. Symmetric noise rate is 20% and 40%, respectively; (c) (d): Encoder is randomly initialized with noise rate 40% and 60%, respectively.

Table 1: Comparing each method on CIFAR10. The model is *learned from scratch without SSL pretraining* for all methods with $\lambda = 1$. Best and last epoch test accuracies are reported: best/last.

| Method | Symm. CIFAR10 | | Asymm. CIFAR10 |
|---|---|---|---|
| | $\varepsilon = 0.6$ | $\varepsilon = 0.8$ | $\varepsilon = 0.4$ |
| CE | 61.29/32.83 | 38.46/15.05 | 67.28/56.6 |
| CE + Regularizer | **69.02/65.13** | **61.94/56.78** | **73.38/58.51** |
| GCE (Zhang & Sabuncu, 2018) | 72.56/62.84 | 40.71/20.53 | 69.19/53.24 |
| GCE + Regularizer | **72.61/68.38** | **63.63/63.05** | **69.79/61.32** |
| FW (Patrini et al., 2017) | 65.95/60.13 | 40.08/26.7 | 68.62/58.01 |
| FW + Regularizer | **68.73/65.90** | **60.94/59.88** | **75.64/67.66** |
| HOC (Zhu et al., 2021b) | 62.53/46.17 | 39.61/16.90 | **85.88**/78.89 |
| HOC + Regularizer | **70.07/66.94** | **60.9/34.90** | 83.53/**82.56** |
| Peer Loss (Liu & Guo, 2020) | 77.52/**76.07** | 15.60/10.00 | **84.47**/68.93 |
| Peer Loss + Regularizer | **77.61**/73.26 | **61.64/53.52** | 81.58/**75.38** |
| ELR (Liu et al., 2020) | 72.56/71.58 | 42.76/23.57 | 86.65/85.27 |
| ELR + Regularizer | **76.95/75.98** | **63.4/57.45** | **88.62/87.92** |

Table 2: Test accuracy for each method on CIFAR10N and CIFAR100N.

| | CE | GCE | Co-Teaching+ | Peer Loss | JoCoR | ELR | CE + Regularizer |
|---|---|---|---|---|---|---|---|
| CIFAR10N (Worst) | 77.69 | 80.66 | 83.26 | 82.53 | 83.37 | 83.58 | **88.74** |
| CIFAR100N | 55.50 | 56.73 | 57.88 | 57.59 | 59.97 | 58.94 | **60.81** |

Table 3: Test accuracy for each method on Clothing1M dataset. All the methods use ResNet50 backbones. DS: Down-Sampling. Reg: With structural regularizer.

| | Foward-T | Co-teaching | CORES+DS | ELR+DS | CE | CE + DS | CE + DS + Reg |
|---|---|---|---|---|---|---|---|
| Initializer | ImageNet | ImageNet | ImageNet | ImageNet | SimCLR | SimCLR | SimCLR |
| Accuracy | 70.83 | 69.21 | 73.24 | 72.87 | 70.90 | 72.95 | **73.48** |

the very beginning and the performance gradually decreases. However, for CE with the regularizer, it helps the network gradually increase the performance as the network reaches the lowest point (over-fitting state). This observation supports Theorem 2 that the regularizer can prevent over-fitting.

Next, we show the regularizer can complement any other loss functions to further improve performance on learning with noisy labels. *I.e.,* we choose $\ell$ in Equation (1) to be other robust losses. The overall experiments are shown in Table 1. It can be observed that our regularizer can complement other loss functions or methods and improve their performance, especially for the last epoch accuracy. Note that we do not apply any tricks when incorporating other losses since we mainly want to observe the effect of the regularizer. It is possible to use other techniques to further improve performance such as multi-model training (Li et al., 2020) or mixup (Zhang et al., 2018a).

**Experiments on real-world label noise** We also test our regularizer on the datasets with real-world label noise: CIFAR10N, CIFAR100N (Wei et al., 2022d) and Clothing1M (Xiao et al., 2015). The results are shown in Table 2 and Table 3. we can find that our regularizer is also effective on the datasets with real-world label noise even when $\ell$ in Equation (1) is simply chosen to be Cross Entropy. More experiments, analyses, and ablation studies can be found in the Appendix.

## 6 CONCLUSIONS

In this paper, we theoretically analyze the memorization effect by showing the relationship among noise rates, sample size, and model capacity. By decoupling DNNs into an encoder followed by a linear classifier, our analyses help reveal the trade-off between fixing or unfixing the encoder during training, which inspires us a new solution to restrict overfitting via representation regularization. Our observations and experiments can serve as a guidance for further research to utilize DNN representations to solve noisy label problems.

ACKNOWLEDGEMENT

This work is partially supported by the National Science Foundation (NSF) under grants IIS-2007951, IIS-2143895, CCF-2023495, and the Office of Naval Research under grant N00014-20-1-22.

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

APPENDIX

**Outline**    The Appendix is arranged as follows: Section A proves Lemmas and Theorems in Section 3. Section B proves Theorem 2 in Section 4 and provides an high level understanding on the regularizer from the perspective of Information Theory. Section C illustrates why down-sampling can decrease the gap of noise rates. Section D provides the effect of distance measure in Eqn (2) ($w = 1$ or 2); ablation study in Section 4; the effect of different SSL pre-trained methods; the performance of shallow network and deeper network on high noise settings; empirical validation of Assumption 2; Experiments towards the regularizer on CIFAR100 dataset. Section E elaborates the detailed experimental setting of all the experiments in the paper.

## A    PROOF FOR LEMMAS AND THEOREMS IN SECTION 3

### A.1    PROOF FOR LEMMA 1

Let $\underline{T} := \min_{X,i} T_{ii}(X)$.

Considering a general instance-dependent label noise where $T_{ij}(X) = \mathbb{P}(\widetilde{Y} = j | Y = i, X)$, we have (Cheng et al., 2021)

$$
\begin{aligned}
&\mathbb{E}_{\widetilde{\mathcal{D}}}[\ell(C(X), \widetilde{Y})] \\
&= \sum_{j \in [K]} \int_x \mathbb{P}(\widetilde{Y} = j, X = x)\ell(C(X), j)\, dx \\
&= \sum_{i \in [K]} \sum_{j \in [K]} \int_x \mathbb{P}(\widetilde{Y} = j, Y = i, X = x)\ell(C(X), j)\, dx \\
&= \sum_{i \in [K]} \sum_{j \in [K]} \mathbb{P}(Y = i) \int_x \mathbb{P}(\widetilde{Y} = j | Y = i, X = x)\mathbb{P}(X = x | Y = i)\ell(C(X), j)\, dx \\
&= \sum_{i \in [K]} \sum_{j \in [K]} \mathbb{P}(Y = i)\mathbb{E}_{\mathcal{D}|Y=i}\left[\mathbb{P}(\widetilde{Y} = j | Y = i, X = x)\ell(C(X), j)\right] \\
&= \sum_{i \in [K]} \sum_{j \in [K]} \mathbb{P}(Y = i)\mathbb{E}_{\mathcal{D}|Y=i}\left[T_{ij}(X)\ell(C(X), j)\right] \\
&= \sum_{i \in [K]} \mathbb{P}(Y = i)\mathbb{E}_{\mathcal{D}|Y=i}\left[T_{ii}(X)\ell(C(X), i)\right] + \sum_{i \in [K]} \sum_{j \in [K], j \neq i} \mathbb{P}(Y = i)\mathbb{E}_{\mathcal{D}|Y=i}\left[T_{ij}(X)\ell(C(X), j)\right] \\
&= \underline{T} \sum_{i \in [K]} \mathbb{P}(Y = i)\mathbb{E}_{\mathcal{D}|Y=i}\left[\ell(C(X), i)\right] + \sum_{i \in [K]} \mathbb{P}(Y = i)\mathbb{E}_{\mathcal{D}|Y=i}\left[(T_{ii}(X) - \underline{T})\ell(C(X), i)\right] \\
&\qquad + \sum_{i \in [K]} \sum_{j \in [K], j \neq i} \mathbb{P}(Y = i)\mathbb{E}_{\mathcal{D}|Y=i}\left[T_{ij}(X)\ell(C(X), j)\right] \\
&= \underline{T}\mathbb{E}_{\mathcal{D}}[\ell(C(X), Y)] + \sum_{j \in [K]} \sum_{i \in [K]} \mathbb{P}(Y = i)\mathbb{E}_{\mathcal{D}|Y=i}[U_{ij}(X)\ell(C(X), j)],
\end{aligned}
$$

where $U_{ij}(X) = T_{ij}(X), \forall i \neq j, U_{jj}(X) = T_{jj}(X) - \underline{T}$.

## A.2 PROOF FOR LEMMA 2

Consider the symmetric label noise. Let $T(X) \equiv T, \forall X$, where $T_{ii} = 1 - \epsilon$, $T_{ij} = \frac{\epsilon}{K-1}, \forall i \neq j$. The general form in Lemma 1 can be simplified as

$$
\mathbb{E}_{\widetilde{\mathcal{D}}}[\ell(C(X), \widetilde{Y})]
$$
$$
= (1 - \epsilon)\mathbb{E}_{\mathcal{D}}[\ell(C(X), Y)] + \frac{\epsilon}{K-1} \sum_{j \in [K]} \sum_{i \in [K], i \neq j} \mathbb{P}(Y = i)\mathbb{E}_{\mathcal{D}|Y=i}[\ell(C(X), j)]
$$
$$
= (1 - \epsilon - \frac{\epsilon}{K-1})\mathbb{E}_{\mathcal{D}}[\ell(C(X), Y)] + \frac{\epsilon}{K-1} \sum_{j \in [K]} \sum_{i \in [K]} \mathbb{P}(Y = i)\mathbb{E}_{\mathcal{D}|Y=i}[\ell(C(X), j)].
$$

When $\ell$ is the 0-1 loss, we have

$$
\sum_{j \in [K]} \sum_{i \in [K]} \mathbb{P}(Y = i)\mathbb{E}_{\mathcal{D}|Y=i}[\ell(C(X), j)] = 1
$$

and

$$
\mathbb{E}_{\widetilde{\mathcal{D}}}[\ell(C(X), \widetilde{Y})] = (1 - \frac{\epsilon K}{K-1})\mathbb{E}_{\mathcal{D}}[\ell(C(X), Y)] + \frac{\epsilon}{K-1}.
$$

Consider the asymmetric label noise. Let $T(X) \equiv T, \forall X$, where $T_{ii} = 1 - \epsilon$, $T_{i,(i+1)_K} = \epsilon$. The general form in Lemma 1 can be simplified as

$$
\mathbb{E}_{\widetilde{\mathcal{D}}}[\ell(C(X), \widetilde{Y})] = (1 - \epsilon)\mathbb{E}_{\mathcal{D}}[\ell(C(X), Y)] + \epsilon \sum_{i \in [K]} \mathbb{P}(Y = i)\mathbb{E}_{\mathcal{D}|Y=i}[\ell(C(X), (i+1)_K)].
$$

## A.3 PROOF FOR THEOREM 1

For symmetric noise, we have:

$$
\mathbb{E}_{\mathcal{D}}\left[\ell(\widehat{C}_{\widetilde{D}}(X), Y)\right] = \frac{\mathbb{E}_{\widetilde{\mathcal{D}}}\left[\ell(\widehat{C}_{\widetilde{D}}(X), \widetilde{Y})\right]}{1 - \epsilon K/(K-1)} - \frac{\epsilon/(K-1)}{1 - \epsilon K/(K-1)}.
$$

Thus the learning error is

$$
\mathbb{E}_{\mathcal{D}}\left[\ell(\widehat{C}_{\widetilde{D}}(X), Y)\right] - \mathbb{E}_{\mathcal{D}}\left[\ell(C_{\mathcal{D}}(X), Y)\right]
$$
$$
= \frac{1}{1 - \epsilon K/(K-1)}\left(\mathbb{E}_{\widetilde{\mathcal{D}}}\left[\ell(\widehat{C}_{\widetilde{D}}(X), \widetilde{Y})\right] - \mathbb{E}_{\widetilde{\mathcal{D}}}\left[\ell(C_{\mathcal{D}}(X), \widetilde{Y})\right]\right).
$$

Let

$$
\hat{\mathbb{E}}_{\widetilde{D}}\left[\ell(C(X), \widetilde{Y})\right] := \frac{1}{N} \sum_{n \in [N]} \ell(C(x_n), \tilde{y}_n).
$$

Noting $\hat{\mathbb{E}}_{\widetilde{D}}\left[\ell(C_{\mathcal{D}}(X), \widetilde{Y})\right] - \hat{\mathbb{E}}_{\widetilde{D}}\left[\ell(\widehat{C}_{\widetilde{D}}(X), \widetilde{Y})\right] \geq 0$, we have the following upper bound:

$$
\mathbb{E}_{\widetilde{\mathcal{D}}}\left[\ell(\widehat{C}_{\widetilde{D}}(X), \widetilde{Y})\right] - \mathbb{E}_{\widetilde{\mathcal{D}}}\left[\ell(C_{\mathcal{D}}(X), \widetilde{Y})\right]
$$
$$
\leq \mathbb{E}_{\widetilde{\mathcal{D}}}\left[\ell(\widehat{C}_{\widetilde{D}}(X), \widetilde{Y})\right] - \hat{\mathbb{E}}_{\widetilde{D}}\left[\ell(\widehat{C}_{\widetilde{D}}(X), \widetilde{Y})\right] + \hat{\mathbb{E}}_{\widetilde{D}}\left[\ell(C_{\mathcal{D}}(X), \widetilde{Y})\right] - \mathbb{E}_{\widetilde{\mathcal{D}}}\left[\ell(C_{\mathcal{D}}(X), \widetilde{Y})\right]
$$
$$
\leq |\mathbb{E}_{\widetilde{\mathcal{D}}}\left[\ell(\widehat{C}_{\widetilde{D}}(X), \widetilde{Y})\right] - \hat{\mathbb{E}}_{\widetilde{D}}\left[\ell(\widehat{C}_{\widetilde{D}}(X), \widetilde{Y})\right]| + |\hat{\mathbb{E}}_{\widetilde{D}}\left[\ell(C_{\mathcal{D}}(X), \widetilde{Y})\right] - \mathbb{E}_{\widetilde{\mathcal{D}}}\left[\ell(C_{\mathcal{D}}(X), \widetilde{Y})\right]|.
$$

Recall $C \in \mathcal{C}$. Denote the VC-dimension of $\mathcal{C}$ by $|\mathcal{C}|$ (Bousquet et al., 2003; Devroye et al., 2013). By Hoeffding inequality with function space $\mathcal{C}$, with probability at least $1 - \delta$, we have

$$
|\mathbb{E}_{\widetilde{\mathcal{D}}}\left[\ell(\widehat{C}_{\widetilde{D}}(X), \widetilde{Y})\right] - \hat{\mathbb{E}}_{\widetilde{D}}\left[\ell(\widehat{C}_{\widetilde{D}}(X), \widetilde{Y})\right]| + |\hat{\mathbb{E}}_{\widetilde{D}}\left[\ell(C_{\mathcal{D}}(X), \widetilde{Y})\right] - \mathbb{E}_{\widetilde{\mathcal{D}}}\left[\ell(C_{\mathcal{D}}(X), \widetilde{Y})\right]|
$$
$$
\leq 2 \arg\max_{C \in \mathcal{C}} |\mathbb{E}_{\widetilde{\mathcal{D}}}\left[\ell(C(X), \widetilde{Y})\right] - \hat{\mathbb{E}}_{\widetilde{D}}\left[\ell(C(X), \widetilde{Y})\right]|
$$
$$
\leq 16 \sqrt{\frac{|\mathcal{C}| \log(N \cdot e/|\mathcal{C}|) + \log(8/\delta)}{2N}}.
$$

Thus

$$\mathbb{E}_{\mathcal{D}}\left[\ell(\widehat{C}_{\widetilde{D}}(X),Y)\right] - \mathbb{E}_{\mathcal{D}}\left[\ell(C_{\mathcal{D}}(X),Y)\right] \le 16\sqrt{\frac{|\mathcal{C}|\log(N\cdot e/|\mathcal{C}|) + \log(8/\delta)}{2N(1-\frac{\epsilon K}{K-1})^2}}.$$

Similarly, for asymmetric noise, we have:

$$\mathbb{E}_{\mathcal{D}}\left[\ell(\widehat{C}_{\widetilde{D}}(X),Y)\right] = \frac{\mathbb{E}_{\widetilde{\mathcal{D}}}\left[\ell(\widehat{C}_{\widetilde{D}}(X),\widetilde{Y}\right]}{1-\epsilon} - \mathsf{Bias}(\widehat{C}_{\widetilde{D}}),$$

where

$$\mathsf{Bias}(\widehat{C}_{\widetilde{D}}) = \frac{\epsilon}{1-\epsilon}\sum_{i\in[K]}\mathbb{P}(Y=i)\mathbb{E}_{\mathcal{D}|Y=i}[\ell(\widehat{C}_{\widetilde{D}}(X),(i+1)_K)].$$

Thus the learning error is

$$\mathbb{E}_{\mathcal{D}}\left[\ell(\widehat{C}_{\widetilde{D}}(X),Y)\right] - \mathbb{E}_{\mathcal{D}}\left[\ell(C_{\mathcal{D}}(X),Y)\right]$$
$$= \frac{1}{1-\epsilon}\left(\mathbb{E}_{\widetilde{\mathcal{D}}}\left[\ell(\widehat{C}_{\widetilde{D}}(X),\widetilde{Y})\right] - \mathbb{E}_{\widetilde{\mathcal{D}}}\left[\ell(C_{\mathcal{D}}(X),\widetilde{Y})\right]\right) + \left(\mathsf{Bias}(C_{\mathcal{D}}) - \mathsf{Bias}(\widehat{C}_{\widetilde{D}})\right)$$

By repeating the derivation for the symmetric noise, we have

$$\mathbb{E}_{\mathcal{D}}\left[\ell(\widehat{C}_{\widetilde{D}}(X),Y)\right] - \mathbb{E}_{\mathcal{D}}\left[\ell(C_{\mathcal{D}}(X),Y)\right] \le 16\sqrt{\frac{|\mathcal{C}|\log(N\cdot e/|\mathcal{C}|) + \log(8/\delta)}{2N}} + \left(\mathsf{Bias}(C_{\mathcal{D}}) - \mathsf{Bias}(\widehat{C}_{\widetilde{D}})\right).$$

### A.4 PROOF FOR COROLLARY 1

**Symmetric noise** Let $\mathcal{C}_1 = \mathcal{G}\circ\mathcal{F}$, $\mathcal{C}_2 = \mathcal{G}|f$. Denote the optimal classifier learned within the above two functions spaces by $C_{\mathcal{D}}^{\mathcal{G}\circ\mathcal{F}}$ and $C_{\mathcal{D}}^{\mathcal{G}|f}$, respectively. Then the approximation errors of both cases can be denoted by $\mathsf{Error}_A(C_{\mathcal{D}}^{\mathcal{G}\circ\mathcal{F}}, C^*)$ and $\mathsf{Error}_A(C_{\mathcal{D}}^{\mathcal{G}|f}, C^*)$. Assume $\mathsf{Error}_A(C_{\mathcal{D}}^{\mathcal{G}\circ\mathcal{F}}, C^*) <$ $\mathsf{Error}_A(C_{\mathcal{D}}^{\mathcal{G}|f}, C^*)$. Note the assumption holds generally and the bias-complexity trade-off does not exist if the assumption does not hold.

From Lemma A.4 in (Shalev-Shwartz & Ben-David, 2014) and our Theorem 1, we know

$$\mathbb{E}|\mathsf{Error}_E(C_{\mathcal{D}}, \widehat{C}_{\widetilde{D}})| \le 16\frac{\sqrt{|\mathcal{C}|\log(4N\cdot e/|\mathcal{C}|)} + 2}{\sqrt{2N}}.$$

Therefore, by requiring the difference between two total generalization errors large than 0, we have:

$$\mathbb{E}_\delta|\Delta_E(\mathcal{C}_1,\varepsilon,\delta)| + \Delta_A(\mathcal{C}_1) - \mathbb{E}_\delta|\Delta_E(\mathcal{C}_2,\varepsilon,\delta)| - \Delta_A(\mathcal{C}_2) \ge 0$$
$$\Leftrightarrow 16\frac{\sqrt{|\mathcal{G}\circ\mathcal{F}|\log(4N\cdot e/|\mathcal{G}\circ\mathcal{F}|)} + 2}{\sqrt{2N(1-\frac{\epsilon K}{K-1})^2}} - 16\frac{\sqrt{|\mathcal{G}|\log(4N\cdot e/|\mathcal{G}|)} + 2}{\sqrt{2N(1-\frac{\epsilon K}{K-1})^2}} + \mathsf{Error}_A(C_{\mathcal{D}}^{\mathcal{G}|f}, C^*) - \mathsf{Error}_A(C_{\mathcal{D}}^{\mathcal{G}\circ\mathcal{F}}, C^*) \ge 0$$
$$\Leftrightarrow 1 - \frac{\epsilon K}{K-1} \le \frac{16}{\sqrt{2N}}\frac{\left(\sqrt{|\mathcal{G}\circ\mathcal{F}|\log(4N\cdot e/|\mathcal{G}\circ\mathcal{F}|)} - \sqrt{|\mathcal{G}|\log(4N\cdot e/|\mathcal{G}|)}\right)}{\mathsf{Error}_A(C_{\mathcal{D}}^{\mathcal{G}|f}, C^*) - \mathsf{Error}_A(C_{\mathcal{D}}^{\mathcal{G}\circ\mathcal{F}}, C^*)}$$

## B  PROOF FOR THEOREMS IN SECTION 4

**Lemma 3.** *If $X$ and $Y$ are independent and follow gaussian distribution: $X \sim \mathcal{N}(\mu_X, \Sigma_X)$ and $Y \sim \mathcal{N}(\mu_Y, \Sigma_Y)$, Then: $\mathbb{E}_{X,Y}(||X-Y||^2) = ||\mu_X - \mu_Y||^2 + tr(\Sigma_X + \Sigma_Y)$.*

### B.1  PROOF FOR THEOREM 2

Before the derivation, we define some notations for better presentation. Following the notation in Section 4, define the labels of $X^{\mathrm{T}}$ as $Y^{\mathrm{T}}$ and the labels of $X^{\mathrm{F}}$ as $Y^{\mathrm{F}}$. Under the label noise, it is easy

to verify $\mathbb{P}(Y^{\text{T}}=1) = \frac{\mathbb{P}(Y=1)\cdot(1-e_+)}{\mathbb{P}(Y=1)\cdot(1-e_+)+\mathbb{P}(Y=0)\cdot(1-e_-)}$ and $\mathbb{P}(Y^{\text{F}}=1) = \frac{\mathbb{P}(Y=0)\cdot e_-}{\mathbb{P}(Y=0)\cdot e_-+\mathbb{P}(Y=1)\cdot e_+}$. Let $p_1 = \mathbb{P}(Y^{\text{T}}=1)$, $p_2 = \mathbb{P}(Y^{\text{F}}=1)$, $g(f(X))$ and $h(f(X))$ to be simplified as $gf(X)$ and $hf(X)$.

In the case of binary classification, $gf(x)$ is one dimensional value which denotes the network prediction on $x$ belonging to $Y=1$. $L_c$ can be written as:

$$\mathbb{E}_{X^{\text{T}},X^{\text{F}}} \underbrace{\left( \frac{||gf(X^{\text{T}})-gf(X^{\text{F}}))||^1}{m_1} - \frac{||hf(X^{\text{T}})-hf(X^{\text{F}})||^2}{m_2}\right)^2}_{\text{denoted as } \Psi(X^{\text{T}},X^{\text{F}})}$$

$$\overset{(a)}{=} \mathbb{E}_{\substack{(X^{\text{T}},Y^{\text{T}})\\(X^{\text{F}},Y^{\text{F}})}}\Psi(X^{\text{T}},X^{\text{F}})$$

$$= p_1 \cdot p_2 \cdot \mathbb{E}_{X^{\text{T}}_+,X^{\text{F}}_+}\Psi(X^{\text{T}}_+,X^{\text{F}}_+) + (1-p_1)\cdot p_2 \cdot \mathbb{E}_{X^{\text{T}}_-,X^{\text{F}}_+}\Psi(X^{\text{T}}_-,X^{\text{F}}_+)$$

$$+ p_1 \cdot (1-p_2)\cdot \mathbb{E}_{X^{\text{T}}_+,X^{\text{F}}_-}\Psi(X^{\text{T}}_+,X^{\text{F}}_-) + (1-p_1)\cdot(1-p_2)\cdot \mathbb{E}_{X^{\text{T}}_-,X^{\text{F}}_-}\Psi(X^{\text{T}}_-,X^{\text{F}}_-)$$

where $m_1$ and $m_2$ are normalization terms from Equation (2). Specifically,

$$m_1 := \lim_{\mathcal{B}\to\infty} \frac{1}{|\mathcal{B}|(|\mathcal{B}|-1)} \sum_{x_n,x_{n'}\in\mathcal{B},n\neq n'} ||gf(x_n)-gf(x'_n)||^1,$$

$$m_2 := \lim_{\mathcal{B}\to\infty} \frac{1}{|\mathcal{B}|(|\mathcal{B}|-1)} \sum_{x_n,x_{n'}\in\mathcal{B},n\neq n'} ||hf(x_n)-hf(x'_n)||^2.$$

(a) is satisfied because $\Psi(X^{\text{T}},X^{\text{F}})$ is irrelevant to the labels. We derive $\Psi(X^{\text{T}}_+,X^{\text{F}}_+)$ as follows:

$$\mathbb{E}_{X^{\text{T}}_+,X^{\text{F}}_+}\Psi(X^{\text{T}}_+,X^{\text{F}}_+)$$

$$\overset{(b)}{=} \mathbb{E}_{X^{\text{T}}_+,X^{\text{F}}_+}\left(\frac{||1-gf(X^{\text{F}}_+)||^1}{m_1} - \frac{||hf(X^{\text{T}}_+)-hf(X^{\text{F}}_+)||^2}{m_2}\right)^2$$

$$\overset{(c)}{=} \mathbb{E}_{X^{\text{T}}_+,X^{\text{F}}_+}\left(\frac{1-gf(X^{\text{F}}_+)}{m_1} - \frac{||hf(X^{\text{T}}_+)-hf(X^{\text{F}}_+)||^2}{m_2}\right)^2$$

$$\overset{(d)}{=} \mathbb{E}_{X^{\text{T}}_+,X^{\text{F}}_+}\left(\frac{gf(X^{\text{F}}_+)}{m_1} - \left(\frac{1}{m_1} - \frac{||hf(X^{\text{T}}_+)-hf(X^{\text{F}}_+)||^2}{m_2}\right)\right)^2$$

(b) is satisfied because from Assumption 1, DNN has confident prediction on clean samples. (c) is satisfied because $gf(X)$ is one dimensional value which ranges from 0 to 1. From Assumption 3, $hf(X_+)$ and $hf(X_-)$ follows gaussian distribution with parameter $(\mu_1,\Sigma)$ and $(\mu_2,\Sigma)$. Thus according to Lemma 3, we have $\mathbb{E}_{X^{\text{T}}_+,X^{\text{F}}_+}||hf(X^{\text{T}}_+)-hf(X^{\text{F}}_+)||^2 = ||\mu_1-\mu_2||^2 + 2\cdot tr(\Sigma)$. Similarly, one can calculate $\mathbb{E}_{X^{\text{T}}_-,X^{\text{F}}_+}||hf(X^{\text{T}}_-)-hf(X^{\text{F}}_+)||^2 = 2\cdot tr(\Sigma)$. It can be seen that (d) is function with respect to $gf(X^{\text{F}}_+)$. Similarly, $\Psi(X^{\text{T}}_-,X^{\text{F}}_+)$ is also a function with respect to $gf(X^{\text{F}}_+)$ while $\Psi(X^{\text{T}}_+,X^{\text{F}}_-)$ and $\Psi(X^{\text{T}}_-,X^{\text{F}}_-)$ are functions with respect to $gf(X^{\text{F}}_-)$. Denote $d(+,+) = \mathbb{E}_{X^{\text{T}}_+,X^{\text{F}}_+}||hf(X^{\text{T}}_+)-hf(X^{\text{F}}_+)||^2$. After organizing $\Psi(X^{\text{T}}_+,X^{\text{F}}_+)$ and $\Psi(X^{\text{T}}_-,X^{\text{F}}_+)$, we have:

$$\min_{gf(X^{\text{F}}_+)} p_1 \cdot p_2 \cdot \mathbb{E}_{X^{\text{T}}_+,X^{\text{F}}_+}\Psi(X^{\text{T}}_+,X^{\text{F}}_+) + (1-p_1)\cdot p_2 \cdot \mathbb{E}_{X^{\text{T}}_-,X^{\text{F}}_+}\Psi(X^{\text{T}}_-,X^{\text{F}}_+)$$

$$\Rightarrow \min_{gf(X^{\text{F}}_+)} \left(\mathbb{E}_{X^{\text{F}}_+}gf(X^{\text{F}}_+)\right)^2 \tag{5}$$

$$- \left(2\cdot p_1(1-\frac{m_1\cdot d(+,+)}{m_2}) + 2\cdot(1-p_1)(\frac{m_1\cdot d(-,+)}{m_2})\right)\cdot \mathbb{E}_{X^{\text{F}}_+}gf(X^{\text{F}}_+)$$

$$+ \text{ constant with respect to } gf(X^{\text{F}}_+)$$

Note in Equation (5), we use $(\mathbb{E}_{X^{\text{F}}_+}gf(X^{\text{F}}_+))^2$ to approximate $\mathbb{E}_{X^{\text{F}}_+}gf(X^{\text{F}}_+)^2$ since from Assumption 2, $\text{var}(g(f(X^{\text{F}}_+))) \to 0$. Now we calculate $m_1$ and $m_2$ from Equation (2):

$$m_1 = p_1 \cdot p_2 \cdot (1 - \mathbb{E}_{X_+^{\mathrm{F}}} gf(X_+^{\mathrm{F}})) + (1 - p_1) \cdot p_2 \cdot \mathbb{E}_{X_+^{\mathrm{F}}} gf(X_+^{\mathrm{F}})$$
$$+ p_1 \cdot (1 - p_2) \cdot (1 - \mathbb{E}_{X_-^{\mathrm{F}}} gf(X_-^{\mathrm{F}})) + (1 - p_1) \cdot (1 - p_2) \cdot \mathbb{E}_{X_-^{\mathrm{F}}} gf(X_-^{\mathrm{F}}) \tag{6}$$

$$m_2 = p_1 \cdot p_2 \cdot d(+,+) + (1 - p_1) \cdot p_2 \cdot d(-,+) + p_1 \cdot (1 - p_2) \cdot d(+,-) + (1 - p_1)(1 - p_2) \cdot d(-,-)$$

Under the condition of $\mathbb{P}(Y = 1) = \mathbb{P}(Y = 0)$, $e_- = e_+$, we have $p_1 = p_2 = \frac{1}{2}$, $m_2 = \frac{4 \cdot tr(\Sigma) + ||\mu_1 - \mu_2||^2}{2}$, $m_1 = \frac{1}{2}$, which is constant with respect to $\mathbb{E}_{X_+^{\mathrm{F}}} gf(X_+^{\mathrm{F}})$ and $\mathbb{E}_{X_-^{\mathrm{F}}} gf(X_-^{\mathrm{F}})$ in Equation (6). Thus Equation (5) is a quadratic equation with respect to $\mathbb{E}_{X_+^{\mathrm{F}}} gf(X_+^{\mathrm{F}})$. Then when Equation (5) achieves global minimum, we have:

$$\mathbb{E}_{X_+^{\mathrm{F}}} gf(X_+^{\mathrm{F}}) = p_1 - \frac{m_1}{m_2}(p_1 \cdot d(+,+) - (1 - p_1) \cdot d(-,+))$$
$$= \frac{1}{2} - \frac{1}{2 + \frac{8 \cdot tr(\Sigma)}{||\mu_1 - \mu_2||^2}} \tag{7}$$

Similarly, organizing $\Psi(X_+^{\mathrm{T}}, X_-^{\mathrm{F}})$ and $\Psi(X_-^{\mathrm{T}}, X_-^{\mathrm{F}})$ gives the solution of $\mathbb{E}_{X_-^{\mathrm{F}}} gf(X_-^{\mathrm{F}})$:

$$\mathbb{E}_{X_-^{\mathrm{F}}} gf(X_-^{\mathrm{F}}) = p_1 + \frac{m_1}{m_2}(p_1 \cdot d(-,-) - (1 - p_1) \cdot d(+,-))$$
$$= \frac{1}{2} + \frac{1}{2 + \frac{8 \cdot tr(\Sigma)}{||\mu_1 - \mu_2||^2}} \tag{8}$$

Denote $\Delta(\Sigma, \mu_1, \mu_2) = 8 \cdot tr(\Sigma)/||\mu_1 - \mu_2||^2$. Now we can write the expected risk as:

$$\mathbb{E}_{\mathcal{D}}\left[\mathbb{1}\left(g(f(X), Y)\right] = (1 - e) \cdot \mathbb{E}_{X^{\mathrm{T}}, Y}\left[\mathbb{1}\left(g(f(X^{\mathrm{T}}), Y)\right] + e \cdot \mathbb{E}_{X^{\mathrm{F}}, Y}\left[\mathbb{1}\left(g(f(X^{\mathrm{F}}), Y)\right]\right.$$
$$\stackrel{(a)}{=} e \cdot \mathbb{E}_{X^{\mathrm{F}}, Y}\left[\mathbb{1}\left(g(f(X^{\mathrm{F}}), Y)\right]\right.$$
$$\stackrel{(b)}{=} e \cdot \left(\frac{1}{2} \cdot \mathbb{E}_{X_+^{\mathrm{F}}, Y=0}\left[\mathbb{1}\left(g(f(X_+^{\mathrm{F}}), 0)\right] + \frac{1}{2} \cdot \mathbb{E}_{X_-^{\mathrm{F}}, Y=1}\left[\mathbb{1}\left(g(f(X_-^{\mathrm{F}}), 1)\right]\right)\right.$$
$$\stackrel{(c)}{=} e \cdot \left(\frac{1}{2} - \frac{1}{2 + \Delta(\Sigma, \mu_1, \mu_2)}\right)$$
$$\tag{9}$$

$(a)$ is satisfied because of Assumption 1 that model can perfectly memorize clean samples. $(b)$ is satisfied because of balanced label and error rate assumption. $(c)$ is satisfied by taking the results from Equation (7) and Equation (8).

Proof Done.

## B.2 HIGH LEVEL UNDERSTANDING ON THE REGULARIZER

Even though we have built Theorem 2 to show SL features can benefit from the structure of SSL features by performing regularization, there still lacks high-level understanding of what the regularization is exactly doing. Here we provide an insight in Theorem 3 which shows the regularization is implicitly maximizing mutual information between SL features and SSL features.

**Theorem 3.** *Suppose there exists a function $\xi$ such that $C(X) = \xi(h(f(X)))$. The mutual information $I(h(f(X)), C(X))$ achieves its maximum when $L_c = 0$ in Eqn. (4),*

The above results facilitate a better understanding on what the regularizer is exactly doing. Note that Mutual Information itself has several popular estimators (Belghazi et al., 2018; Hjelm et al., 2018). It is a very interesting future direction to develop regularizes based on MI to perform regularization by utilizing SSL features.

**Proof for Theorem 3**: We first refer to a property of Mutual Information:

$$I(X;Y) = I(\psi(X); \phi(Y)) \tag{10}$$

where $\psi$ and $\phi$ are any invertible functions. This property shows that mutual information is invariant to invertible transformations (Cover, 1999). Thus to prove the theorem, we only need to prove that $\xi$ in Theorem 3 must be an invertible function when Equation (4) is minimized to 0. Since when $\xi$ is invertible, $I(h(f(X)), C(X)) = I(h(f(X)), \xi(h(f(X)))) = I(h(f(X)), h(f(X)))$.

We prove this by contradiction.

Let $t_i = h(f(x_i))$ and $s_i = g(f(x_i))$. Suppose $\xi$ is not invertible, then there must exists $s_i$ and $s_j$ where $s_i \neq s_j$ which satisfy $t_j = \xi(s_i) = t_i$. However, under this condition, $t_i - t_j = 0$ and $s_i - s_j \neq 0$, Equation (4) can not be minimized to 0. Thus when Equation (4) is minimized to 0, $\xi$ must be an invertible function.

Proof done.

### B.3 PROOF FOR LEMMA 3

By the independence condition, $Z = X - Y$ also follows gaussian distribution with parameter $(\mu_X - \mu_Y, \Sigma_X + \Sigma_Y)$.

Write $Z$ as $Z = \mu + LU$ where $U$ is a standard gaussian and $\mu = \mu_X - \mu_Y$, $LL^T = \Sigma_X + \Sigma_Y$. Thus

$$||Z||^2 = Z^T Z = \mu^T \mu + \mu^T LU + U^T L^T \mu + U^T L^T LU \tag{11}$$

Since $U$ is standard gaussian, $\mathbb{E}(U) = \mathbf{0}$. We have

$$
\begin{aligned}
\mathbb{E}(||Z||^2) &= \mu^T \mu + \mathbb{E}(U^T L^T LU) \\
&= \mu^T \mu + \mathbb{E}(\sum_{k,l} (L^T L)_{k,l} U_k U_l) \\
&\overset{(a)}{=} \mu^T \mu + \sum_k (L^T L)_{k,k} \\
&= \mu^T \mu + tr(L^T L) \\
&= ||\mu_X - \mu_Y||^2 + tr(\Sigma_X + \Sigma_Y)
\end{aligned}
\tag{12}
$$

(a) is satisfied because $U$ is standard gaussian, thus $\mathbb{E}(U_k^2) = 1$ and $\mathbb{E}(U_k U_l) = 0$ $(k \neq l)$.

Proof Done.

## C  ILLUSTRATING DOWN-SAMPLING STRATEGY

We illustrate in the case of binary classification with $e_+ + e_- < 1$. Suppose the dataset is balanced, at the initial state, $e_+ > e_-$. After down-sampling, the noise rate becomes $e_+^*$ and $e_-^*$. We aim to prove two propositions:

**Proposition 1.** *If $e_+$ and $e_-$ are known, the optimal down-sampling rate can be calculated by $e_+$ and $e_-$ to make $e_+^* = e_-^*$*

**Proposition 2.** *If $e_+$ and $e_-$ are not known. When down-sampling strategy is to make $\mathbb{P}(\widetilde{Y} = 1) = \mathbb{P}(\widetilde{Y} = 0)$, then $0 < e_+^* - e_-^* < e_+ - e_-$.*

*Proof for Proposition 1:* Since dataset is balanced with initial $e_+ > e_-$, we have $\mathbb{P}(\widetilde{Y} = 1) < \mathbb{P}(\widetilde{Y} = 0)$. Thus down-sampling is conducted at samples whose observed label are 0. Suppose the random down-sampling rate is $r$, then $e_+^* = \frac{r \cdot e_+}{1 - e_+ + r \cdot e_+}$ and $e_-^* = \frac{e_-}{r \cdot (1 - e_-) + e_-}$. If $e_+^* = e_-^*$, we have:

$$\frac{r \cdot e_+}{1 - e_+ + r \cdot e_+} = \frac{e_-}{r \cdot (1 - e_-) + e_-} \tag{13}$$

Thus the optimal down-sampling rate $r = \sqrt{\frac{e_- \cdot (1-e_+)}{e_+ \cdot (1-e_-)}}$, which can be calculated if $e_-$ and $e_+$ are known.

*Proof for Proposition 2:* If down sampling strategy is to make $\mathbb{P}(\widetilde{Y} = 1) = \mathbb{P}(\widetilde{Y} = 0)$, then $r \cdot (e_+ + 1 - e_-) = 1 - e_+ + e_-$, we have $r = \frac{1 - e_+ + e_-}{1 - e_- + e_+}$. Thus $e_+^*$ can be calculated as:

$$e_+^* = \frac{r \cdot e_+}{1 - e_+ + r \cdot e_+}$$

$$= \frac{(1 - e_+ + e_-) \cdot e_+}{(1 - e_+) \cdot (1 - e_- + e_+) + e_+ \cdot (1 - e_+ + e_-)}$$

Denote $\alpha = \frac{1 - e_+ + e_-}{(1-e_+) \cdot (1 - e_- + e_+) + e_+ \cdot (1 - e_+ + e_-)}$. Since $e_+ > e_-$, $1 - e_- + e_+ > 1 - e_+ + e_-$, $\alpha = \frac{1 - e_+ + e_-}{(1-e_+) \cdot (1 - e_- + e_+) + e_+ \cdot (1 - e_+ + e_-)} < \frac{1 - e_+ + e_-}{(1-e_+) \cdot (1 - e_+ + e_-) + e_+ \cdot (1 - e_+ + e_-)} = 1$.

Similarly, $e_-^*$ can be calculated as:

$$e_-^* = \frac{e_-}{e_- + r \cdot (1 - e_-)}$$

$$= \frac{(1 - e_- + e_+) \cdot e_-}{e_- \cdot (1 - e_- + e_+) + (1 - e_-) \cdot (1 - e_+ + e_-)}$$

Denote $\beta = \frac{1 - e_- + e_+}{e_- \cdot (1 - e_- + e_+) + (1 - e_-) \cdot (1 - e_+ + e_-)}$. Since $e_+ > e_-$, $1 - e_- + e_+ > 1 - e_+ + e_-$, $\beta = \frac{1 - e_- + e_+}{e_- \cdot (1 - e_- + e_+) + (1 - e_-) \cdot (1 - e_+ + e_-)} > \frac{1 - e_- + e_+}{e_- \cdot (1 - e_- + e_+) + (1 - e_-) \cdot (1 - e_- + e_+)} = 1$. Since $\alpha \cdot e_+ < e_+$ and $\beta \cdot e_- > e_-$, we have $e_+^* - e_-^* = \alpha \cdot e_+ - \beta \cdot e_- < e_+ - e_-$.

Next, we prove $e_+^* > e_-^*$, following the derivation below:

$$e_+^* > e_-^*$$
$$\implies \frac{r \cdot e_+}{1 - e_+ + r \cdot e_+} > \frac{e_-}{e_- + r \cdot (1 - e_-)}$$
$$\implies r > \sqrt{\frac{e_- \cdot (1 - e_+)}{e_+ \cdot (1 - e_-)}} \tag{14}$$
$$\implies \frac{1 - e_+ + e_-}{1 - e_- + e_+} > \sqrt{\frac{e_- \cdot (1 - e_+)}{e_+ \cdot (1 - e_-)}}$$
$$\implies e_+ \cdot (1 - e_+) + \frac{e_+ \cdot e_-^2}{1 - e_+} > e_- \cdot (1 - e_-) + \frac{e_- \cdot e_+^2}{1 - e_-}$$

Let $f(e_+) = e_+ \cdot (1 - e_+) + \frac{e_+ \cdot e_-^2}{1 - e_+} - e_- \cdot (1 - e_-) - \frac{e_- \cdot e_+^2}{1 - e_-}$. Since we have assumed $e_- < e_+$ and $e_- + e_+ < 1$. Thus proving $e_+^* > e_-^*$ is identical to prove $f(e_+) > 0$ when $e_- < e_+ < 1 - e_-$.

Firstly, it is easy to verify when $e_+ = e_-$ or $e_+ = 1 - e_-$, $f(e_+) = 0$. From Mean Value Theory, there must exists a point $e_0$ which satisfy $f'(e_0) = 0$ where $e_+ < e_0 < 1 - e_-$. Next, we differentiate $f(e_+)$ as follows:

$$f'(e_+) = \frac{(1 - e_+)^2 \cdot (1 - e_-) + e_-^2 \cdot (1 - e_-) - 2 \cdot e_+ (1 - e_+)^2}{(1 - e_+)^2 \cdot (1 - e_-)} \tag{15}$$

It can be verified that $f'(e_-) = \frac{1 - e_-}{(1 - e_-)^2 \cdot (1 - e_-)} > 0$ and $f'(1 - e_-) = \frac{0}{e_-^2 \cdot (1 - e_-)} = 0$.

Further differentiate $f'(e_+)$, we get when $e_+ < 1 - ((1 - e_-) \cdot e_-^2)^{\frac{1}{3}}$, $f''(e_+) < 0$ and when $e_+ > 1 - ((1 - e_-) \cdot e_-^2)^{\frac{1}{3}}$, $f''(e_+) > 0$. Since $e_- < e_+$ and $e_- + e_+ < 1$, we have $e_- < \frac{1}{2}$ and $e_- < 1 - ((1 - e_-) \cdot e_-^2)^{\frac{1}{3}} < 1 - e_-$, *i.e.*, $1 - ((1 - e_-) \cdot e_-^2)^{\frac{1}{3}}$ locates in the point between $e_-$ and $1 - e_-$. Thus, when $e_- < e_+ < 1 - ((1 - e_-) \cdot e_-^2)^{\frac{1}{3}}$, $f(e_+)$ is a strictly concave function and when $1 - ((1 - e_-) \cdot e_-^2)^{\frac{1}{3}} < e_+ < 1 - e_-$, $f(e_+)$ is a strictly convex function.

Since $f'(e_-) > 0$ and $f'(1 - e_-) = 0$, $e_0$ must locates in the point between $e_-$ and $1 - ((1 - e_-) \cdot e_-^2)^{\frac{1}{3}}$ which satisfy $f'(e_0) = 0$. Thus when $e_- < e_+ < e_0$, $f(e_+)$ monotonically increases and when

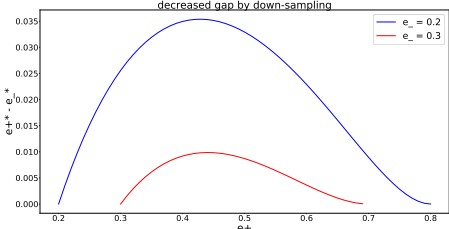

Figure 7: Visualizing decreased gap by down-sampling strategy.

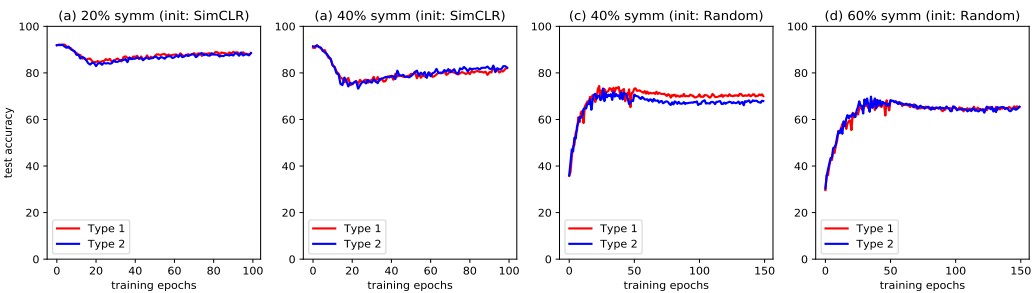

Figure 8: Comparing difference choices of distance measure in Equation (2). Type 1 denotes using $l_2$ norm to calculate distance between SL features and square $l_2$ norm to calculate distance between SSL features, which is adopted in our paper. Type 2 denotes using $l_2$ norm to calculate distance for both SL and SSL features.

$e_0 < e_+ < 1 - e_-$, $f(e_+)$ monotonically decreases. Since $f(e_-) = f(1 - e_-) = 0$. We have $f(e_+) > 0$ when $e_- < e_+ < 1 - e_-$.

Proof done.

We depict a figure in Figure 7 to better show the effect of down-sampling strategy. It can be seen the curves in the figure well support our proposition and proof. When $e_+ - e_-$ is large, down-sampling strategy to make $\mathbb{P}(\widetilde{Y} = 1) = \mathbb{P}(\widetilde{Y} = 0)$ can well decrease the gap even we do not know the true value of $e_-$ and $e_+$.

## D  MORE DISCUSSIONS AND EXPERIMENTS

### D.1  THE EFFECT OF DISTANCE MEASURE IN EQN (2)

In this paper and experiment, we use $l_2$ norm to calculate the feature distance between SL features and square $l_2$ norm to calculate the distance between SSL features. This choice can lead to good performance from Theory 2 and Figure 6. Practically, since structure regularization mainly captures the relations, different choice does not make a big effect on the performance. We perform an experiment in Figure 8 which shows that the performance of both types are quite close.

### D.2  ABLATION STUDY

In Figure 3, SSL training is to provide SSL features to regularize the output of linear classifier $g$. However, SSL training itself may have a positive effect on DNN. To show the robustness mainly comes from the regularizer rather than SSL training, we perform an ablation study in Figure 9. From the experiments, it is the regularizer that alleviates over-fitting problem of DNN.

### D.3  THE EFFECT OF DIFFERENT SSL-PRETRAINED METHODS

Our experiments are not restricted to any specific SSL method. Experimentally, other SSL methods are also adoptable to pre-train SSL encoders. In Figure 5, SimCLR (Chen et al., 2020) is adopted

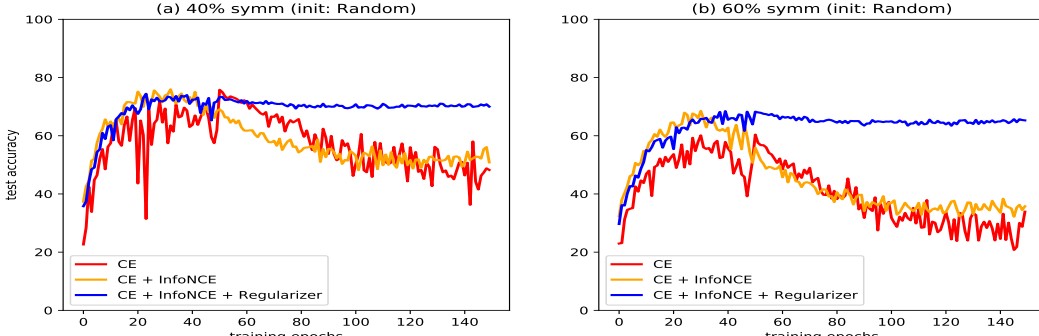

Figure 9: Ablation study of using the regularizer to train DNN on noisy dataset.

to pre-train SSL encoder. For a comparison, we pre-train a encoder with Moco on CIFAR10 and fine-tune linear classifier on noisy labels in Table 4.

Table 4: Comparing different SSL methods on CIFAR10 with symmetric label noise

| Method | Symm label noise ratio | | | |
|---|---|---|---|---|
| | 0.2 | 0.4 | 0.6 | 0.8 |
| CE (fixed encoder with SimCLR init) | 91.06 | 90.73 | 90.2 | 88.24 |
| CE (fixed encoder with MoCo init) | 91.55 | 91.12 | 90.45 | 88.51 |

It can be observed that different SSL methods have very similar results.

### D.4 Performance of shallow network and deeper network on high noise settings.

Apart from empirical results in Section 5, we also provide empirical evidence that a shallow network performs better than deeper network on high-symmetric noise settings to further validate Corollary 1. Since similar to fixing the encoder, a shallow network also has lower capacity than deeper network. The experimental settings are as follows: network structure: ResNet18 vs ResNet50, dataset: CIFAR-10, loss: Cross entropy; number of epochs (100), batch size (64), learning rate (0.1 at first 50 epochs and 0.01 for last 50 epochs), optimizer (SGD). We report the best epoch test accuracy in Table 5 :

Table 5: Comparing the performance of different network structures on CIFAR-10 with symmetric label noise

| Models | Symm label noise ratio | |
|---|---|---|
| | 0.8 | 0.85 |
| ResNet18 | 45.57 | 33.06 |
| ResNet50 | 38.74 | 30.54 |

It can be observed that a shallow network behaves better for high symmetric noise ratio which supports our claim in the paper.

### D.5 Validation of Assumption 2

The zero variance assumption (Assumption 2) to proceed the proof in Theorem 2 is backed up by (Zhang et al., 2016) showing that DNN will memorize all the noisy samples when DNN converges,

resulting to near 0 loss. We perform experiments on CIFAR-100 with symmetric label noise to validate this assumption. The results are reported in Table 6. It can be seen that the variance of noisy

Table 6: Variance of each noisy id in CIFAR-100 with training epochs

| Labels id | Epoch 50 | Epoch 100 | Epoch 150 | Epoch 200 |
|-----------|----------|-----------|-----------|-----------|
| 0 | 3.829 | 0.038 | 0.009 | 0.001 |
| 1 | 4.153 | 0.019 | 0.006 | 0.005 |
| 2 | 3.388 | 0.055 | 0.015 | 0.002 |
| 3 | 3.572 | 0.018 | 0.0002 | 0.0002 |
| 4 | 3.952 | 0.112 | 0.002 | 0.0007 |

samples in each label id is tending to 0 when training converges.

## D.6 Validation of the effect of batch size

We perform experiments on CIFAR100 under symmetric label noise ratio 0.6 with our regularizer for different batch size. Table 7 shows that increasing batch size has slight perfomance gain.

## D.7 Experiments towards regularizer on CIFAR100

In this section, we examine our regularizer on CIFAR100 dataset with certain SOTA methods from (Liu et al., 2020). Results are reported in Table 8 from which we can see that our proposed regularizer can also improve performance on CIFAR100 dataset.

## E Detailed setting of experiments

**Datasets:** We use DogCat, CIFAR10, CIFAR100, CIFAR10N and CIFAR100N and Clothing1M for experiments. DogCat has 25000 images. We randomly choose 24000 images for training and 1000 images for testing. For CIFAR10 and CIFAR100, we follow standard setting that use 50000 images for training and 10000 images for testing. CIFAR10N and CIFAR100N have the same images of CIFAR10 and CIFAR100 except the labels are annotated by real human via Amazon Mturk which contains real-world huamn noise. For Clothing1M, we use noisy data for training and clean data for testing.

**Setting for Figure 1:** We use ResNet34 for conducting the experiments. All the experiments in Figure 1 are trained from scratch with hyper-parameters below: learning rate (0.1 at first 50 epochs and 0.01 for last 50 epochs), batchsize (256), optimizer (SGD).

**Setting in Section 5.1 (Figure 5 and Figure 4):** SimCLR is deployed for SSL pre-training with ResNet50 for DogCat and ResNet34 for CIFAR10 and CIFAR100. Each model is pre-trained by 1000 epochs with Adam optimizer (lr = 1e-3) and batch-size is set to be 512. During fine-tuning, we fine-tune the classifier on noisy dataset with Adam (lr = 1e-3) for 100 epochs and batch-size is set to be 256.

**Setting in Section 5.2:** For Table 1, all the methods are trained from scratch with learning rate set to be 0.1 at the initial state and decayed by 0.1 at 50 epochs. For Table 2 and Table 3, the encoder is pre-trained by SimCLR and we finetune the encoder on the noisy dataset with CE + Regularier. The optimizer is Adam with learning rate 1e-3 and batch-size 256. Note that in Eqn (4), we use MSE loss for measuring the relations between SL features and SSL features. However, since MSE loss may cause gradient exploration when prediction is far from ground-truth, we use smooth $l_1$ loss instead. Smooth $l_1$ loss is an enhanced version of MSE loss. When prediction is not very far from ground-truth, smooth $l_1$ loss is MSE, and MAE when prediction is far.

**Setting for Table 8**: All the methods are trained from scratch with learning rate 0.001. The optimizer is Adam and the training epochs is 100. Note when applying regularizer with each method, for example, Regularizer + CE, we first use CE to warmup DNN for certain epochs, then apply regularizer to prevent overfitting.

Table 7: Comparison of test accuracy with different batch size on CIAFR100 under symmetric noise ratio 0.6.

| Methods | batch size 100 | batch size 200 | batch size 300 |
|---|---|---|---|
| CE + Regularizer | 39.08 | 40.07 | 40.14 |

Table 8: Comparison of test accuracies with each method on CIFAR100. The model is *learned from scratch* for all methods with $\lambda = 1$. Best and last epoch accuracies are reported: best/last.

| Method | Symm. CIFAR100 | | Asymm. CIFAR100 |
|---|---|---|---|
| | $\varepsilon = 0.6$ | $\varepsilon = 0.8$ | $\varepsilon = 0.4$ |
| ELR (Liu et al., 2020) | 34.36/14.21 | 18.95/5.54 | 55.36/33.5 |
| ELR + Regularizer | **37.17/23.45** | **20.92/13.16** | **56.55/36.49** |
| Bootstrap (Reed et al., 2014) | 34.21/13.8 | 18.45/6.68 | 39.28/30.93 |
| Bootstrap + Regularizer | **35.63/27.88** | **19.92/16.3** | **40.21/34.56** |
| FW (Patrini et al., 2017) | 38.72/22.26 | 17.29/8.15 | 42.35/**40.46** |
| FW + Regularizer | **39.01/29.39** | **26.92/21.02** | **43.35**/40.02 |
| SL (Wang et al., 2019) | 34.16/14.12 | 19.17/**6.68** | 40.76/32.07 |
| SL + Regularizer | **34.20/15.3** | **19.65**/6.02 | **40.81/34.05** |
| GCE (Liu et al., 2020) | 38.08/26.79 | 23.24/16.9 | 41.26/31.4 |
| GCE + Regularizer | **44.27/43.48** | **35.01/34.55** | **42.35/35.74** |

