# OpenReview forum: "Mitigating Memorization of Noisy Labels via Regularization between Representations"
_ICLR.cc/2023/Conference — ICLR 2023 poster_

### Official Review · Reviewer_wiWY · 2022-10-16

**Confidence:** 5
**Correctness:** 3
**Technical Novelty And Significance:** 3
**Empirical Novelty And Significance:** 3
**Recommendation:** 8

**Clarity, Quality, Novelty And Reproducibility:**

**Clarity.** The clarity could be improved to further enhance this paper. See the above questions.

**Quality.** The quality of this paper is great for me.

**Novelty.** The conceptual novelty is a bit limited. The technical novelty is great for me.

**Reproducibility.** The reproducibility is good.

**Strength And Weaknesses:**

**Strengths**
1. The paper theoretically reveals how the noise rate affects traditional learning with noisy labels. Besides, the theoretical bounds are connected to the model capacity, which motivates the design of their framework.

2. The proposed framework is demonstrated to be effective from both the theoretical and experimental aspects. Specifically, the theoretical analyses show why and how the self-supervised features can help regularize the learning with noisy labels (weak supervision), which provides insights for understanding the benefit of self-supervised learning in making the model robust to noisy labels.

**Weaknesses and Questions**
1. The paper claims that "early stopping will handle overfitting wrong labels at the cost of underfitting clean samples if not tuned properly". However, the issue may be also not addressed by this paper. Perhaps, holistic regularization cannot avoid the underfitting issue well.

2. It is not very new to exploit contrastive learning to handle noisy labels, e.g., [1] and [2]. More discussions about related work can be added.

3. It seems that Figure 2 is not highly related to the proposed method of this paper. Besides, the power of reducing model capacities of three different paths is different. Could they always converge the same classifier?

4. The analysis of the approximation error is not very solid. Besides, the trade-off analysis may contradict the double descent phenomenon in deep learning. How to address this concern?

5. It is a bit hard to follow "a larger $\mathcal{C}$ will lead to smaller approximation error but at the cost of larger estimation error given large $N$". It seems that the paper provides an error bound, but not the exact values.

6. Assumption 2 is a bit to understand since different classes always have different memorization rates in training [3].

7. Theorem 2 is not highly related to the proposed method/framework. It seems that the theorem tells us the power of the Gaussian assumption in tackling noisy labels, which is similar to [4].

8. From Figure 5, subfigures (a)--(c) show the crossing points between fixed and unfixed encoders are different, i.e., the noise rates are 0.0, 0.4, and 0.7 for different datasets. The authors should provide some insights into this observation.

9. The authors claim the performance of a fixed encoder depends on the pre-trained model. However, current experiments only focus on SimCLR. It is interesting to see whether the proposed framework works well with other self-supervised learning methods.

10. It is interesting to see whether the disentangled representations would be beneficial.

11. Minor comments:
- The name of Section 3 could be revised. The memorization effect mainly claims that the deep network would first fit training data with clean labels and incorrect labels. The analysis and discussion are more related to the memorization and impacts of mislabeled data.
- The citation format needs to be revised. Some ''\cite{}'' need to be changed to ''\citep{}''.
- It is interesting to show the effect of batch size on the effectiveness of the regularizer since the regularizer is based on self-supervised learning.
----
[1] Shikun Li, et al. Selective-Supervised Contrastive Learning with Noisy Labels. CVPR 2022.
[2] Diego Ortego, et al. Multi-objective Interpolation Training for Robustness to Label Noise. CVPR 2021.
[3] Yiwen Wang, et al. Symmetric Cross Entropy for Robust Learning with Noisy Labels. ICCV 2019.
[4] Kimin Lee, et al. Robust Inference via Generative Classifiers for Handling Noisy Labels. ICML 2019.

**Summary Of The Paper:**

This paper discusses the memorization effect of deep neural networks and proposes a framework to mitigate the negative effects of memorizing wrong information when learning with noisy labels. The analyses focus on the tradeoff between estimation error and approximation error due to the model capacity. By two case studies, learning with a fixed encoder and learning with an unfixed encoder, the authors conclude that restricting the search space by fixing the encoder reduces the estimation error but possibly increases approximation errors. Based on the takeaways from the tradeoff analyses, the authors further propose a learning framework that tries to combine the beneficial parts of fixing the encoder (low estimation error) and not fixing the encoder (low approximation error). Specifically, they exploit the power of self-supervision that regularizes supervised learning with self-supervised features. The advantage of the proposed framework is also theoretically analyzed in a simplified case. They also run experiments on both the synthetic label noise and real-world label noise to demonstrate the effectiveness of their framework.

**Summary Of The Review:**

This paper presents work in learning with noisy labels. Both theoretical analysis and experimental results can provide insights into the community. The reviewer affirms its worth. The responses to the mentioned concerns are expected.

####Post rebuttal####

The response is detailed and addresses my concerns. I hence raise my score.

---

> ### Author Response · Authors · 2022-11-13
> **Response to Reviewer wiWY (part 1)**
>
> Thanks for your detailed comments and suggestions. Our responses for your questions are as follows:
>
> **Question 1:** The paper claims that "early stopping will handle overfitting wrong labels at the cost of underfitting clean samples if not tuned properly". However, the issue may be also not addressed by this paper. Perhaps, holistic regularization cannot avoid the underfitting issue well.
>
> **Response 1:** We admit that holistic regularization may not avoid the underfitting problems. Similar phenomenon is observed by [R1] that shows that MAE loss [R2] can address overfitting but has severe underfitting problem. To evaluate whether a loss/regularization/framework can both address overfitting and underfitting well, best epoch accuracy and last epoch accuracy are all reported along the training procedure. Thus in Table 1, we report both two evaluations to show the benefit of our regularizer.
>
> [R1] Generalized Cross Entropy Loss for Training Deep Neural Networks with Noisy Labels. NeurlPS 2018.
>
> [R2] Robust Loss Functions under Label Noise for Deep Neural Networks. AAAI 2017.
>
> **Question 2:** It is not very new to exploit contrastive learning to handle noisy labels, e.g., [R1] and [R2]. More discussions about related work can be added.
>
> **Response 2:** Thank you for bringing there two works to our attention. [R3] selects confident clean sample pairs based on the Supervised Contrastive features for training a robust classifier. [R4] also uses Supervised Contrastive learning, along with a interpolation-based semi-supervised loss for learning robust classifier. Different from these two works, we propose a self-supervised-based regularizer to correct supervised features and provide theoreitcal guarantee (Theorem 2). We will cite and discuss more papers in the related work.
>
>
> [R3]: Selective-Supervised Contrastive Learning with Noisy Labels. CVPR 2022.
>
> [R4]: Multi-objective Interpolation Training for Robustness to Label Noise. CVPR 2021.
>
> **Question 3:** It seems that Figure 2 is not highly related to the proposed method of this paper. Besides, the power of reducing model capacities of three different paths is different. Could they always converge the same classifier?
>
> **Response 3:** Figure 2 is to show possible ways (learning paths) for learning a robust classifier. Different learning path has various effects on label noise types (Theorem 1, Corollary 1). The analyses of generalization error of these learning paths motivates us to propose a regularizer to compromise between fixed and unfixed encoder (\textbf{insights and takeaways in Section 3}).
>
> In the most cases, these learning paths do not converge to the same classifier. For example, from Corollary 1, when the condition holds, fixed encoder (Path-3 in Figure 2) has lower generalization error then unfixed encoder (Path-2 in Figure 2).
>
> **Question 4:** The analysis of the approximation error is not very solid. Besides, the trade-off analysis may contradict the double descent phenomenon in deep learning. How to address this concern? \& It is a bit hard to follow "a larger C will lead to smaller approximation error but at the cost of larger estimation error given large
> N. It seems that the paper provides an error bound, but not the exact values.
>
> **Response 4:** We are aware that a tight generalization bound for an arbitrary neural network is complicated and an
> open question. We did not get into it and just use a traditional term [R5] to summarize the possible effect
> of networks with different capacity. Thus Corollary is derived as a error bound. However, the experiments in Figure 4 and Figure 5 supports our analyses very well.
>
> The setting of double descent phenomenon (Figure 1 in [R6]) is fixing the depth and varying the width, which is different from [R7] that observes the approximation error decreases as the depth increases. Our analyses and settings in the paper are concentrated on the depth rather than width. For example, Table 5 in D.4, supplementary material also supports our analyses in Section 3 that shallow (low capacity) model performs better than deeper (large capacity) model for high symmetric label noise. We will make it more clear in the revised version.
>
>
> [R5] Understanding machine learning: From theory to algorithms. Cambridge university press, 2014.
>
> [R6] Deep Double Descent: Where Bigger Models and More Data Hurt. ICLR 2020.
>
> [R7] Deep Residual Learning for Image Recognition. CVPR 2016.

---

> ### Author Response · Authors · 2022-11-13
> **Response to Reviewer wiWY (part 2)**
>
> **Question 5:** Assumption 2 is a bit to understand since different classes always have different memorization rates in training [R8].
>
> **Response 5:** Theorem 2, which uses the zero variance assumption (Assumption 2), shows the \textbf{optimal (convergence) state} when applying the regularizer.  The zero variance is validated in Table 6 (Appendix D.5, page 23). It shows that the variance of noisy samples in each label id is close to 0 when network converges.
>
> [R8] Symmetric Cross Entropy for Robust Learning with Noisy Labels. ICCV 2019.
>
> **Question 6:** Theorem 2 is not highly related to the proposed method/framework. It seems that the theorem tells us the power of the Gaussian assumption in tackling noisy labels, which is similar to [R9].
>
> **Response 6:** We use Gaussian assumption for self-supervised features mainly for finding a closed-form solution in Theorem 2. Additionally, if we understand correctly, the only similar point between ours and [R9] is the Gaussian assumption. Please note some literatures also observe that self-supervised features tend to follow Gaussian distribution [R10]. As we mentioned in the paragraph below Assumption 3, we use Gaussian due to its simplicity and it can provide us a with a closed-form solution. Other reasonable distribution can also be assumed. We care more about the insights provided by Theorem 2 rather than a fancy bound. Specifically, Theorem 2 is to show the effectiveness of the regularizer, I.e., $L_{c}$ in Equation (4), which implies that ideally, if the SSL features from different classes are well separated (large $||u_1 - u_2||$ and low $\Sigma$ in Assumption 3), then DNN has very low generalization error or overfitting effect.
>
> [R9] Robust Inference via Generative Classifiers for Handling Noisy Labels. ICML 2019.
>
> [R10]: Understanding contrastive representation learning through alignment and uniformity on the hypersphere. ICML 2020
>
> **Question 7:** From Figure 5, subfigures (a)--(c) show the crossing points between fixed and unfixed encoders are different, i.e., the noise rates are 0.0, 0.4, and 0.7 for different datasets. The authors should provide some insights into this observation.
>
> **Response 7:** Good comment. This observation can be explained by Corollary 1.  Corollary 1 implies that if the encoder is learned very well, i.e., $\textsf{Error}\_{A}(C\_{\mathcal D}^{\mathcal G \circ \mathcal F},C^*) \approx  \textsf{Error}\_{A}(C\_{\mathcal D}^{\mathcal G | f},C^*)$, fixing the encoder has benefits over unfixed encoder even when noise rate is small. In Figure 5, since for DogCat, CIFAR10 and CIFAR100 dataset, each class have 12500 samples, 5000 samples and 500 samples, respectively. When applying the self-supervised learning on these datasets, the encoder quality is $\text{DogCat} > \text{CIFAR10} > \text{CIFAR100}$. Thus the crossing point is small for DogCat and large for CIFAR100.
>
> **Question 8:** The authors claim the performance of a fixed encoder depends on the pre-trained model. However, current experiments only focus on SimCLR. It is interesting to see whether the proposed framework works well with other self-supervised learning methods.
>
> **Response 8:** We have provide experiments based on MoCo [R11] in Table 4 in the supplementary material. We find the experiments on MoCo also support our claim.
>
> [R11] Momentum Contrast for Unsupervised Visual Representation Learning. CVPR 2020.

---

> ### Author Response · Authors · 2022-11-13
> **Response to Reviewer wiWY (part 3)**
>
> **Question 9:** It is interesting to see whether the disentangled representations would be beneficial.
>
> **Response 9:** Good suggestion. Our paper is mostly based on SimCLR. For disentangled representation, we use the algorithm IPIRM from [R12] and use their released code for pre-training the encoder. The following Table shows different approaches on CIFAR100 with varied noise ratio for instance dependent label noise. It can be observed that self-supervised disentangled representation even shows better performance than vanilla SimCLR.
>
> | Method           | inst 0.3 | inst 0.4 | inst 0.5 | inst 0.6 |
> | ---------------- | :------: | :------: | -------- | -------- |
> | CE (random init) |  43.57   |  35.17   | 27.07    | 18.25    |
> | CE (SimCLR init) |  58.95   |   49.7   | 36.87    | 25.07    |
> | CE (IPIRM init)  |  64.92   |  56.18   | 43.75    | 30.36    |
>
> [R12] Self-supervised learning disentangled group representation as feature. NeurlPS 2021.
>
> **Question 10:** The name of Section 3 could be revised. The memorization effect mainly claims that the deep network would first fit training data with clean labels and incorrect labels. The analysis and discussion are more related to the memorization and impacts of mislabeled data.
>
> **Response 10:** Thanks for the suggestions. We have renamed the title of Section 3.
>
> **Question 11:** The citation format needs to be revised. Some cite need to be changed to citep.
>
> **Response 11:** Thanks for pointing out. We have fixed this issue.
>
> **Question 12:** It is interesting to show the effect of batch size on the effectiveness of the regularizer since the regularizer is based on self-supervised learning.
>
> **Response 12:** We perform experiments on CIFAR100 under symmetric label noise ratio 0.6 with our regularizer for different batch size. The following Table shows that increasing batch size has slight perfomance gain.
>
> | Method            | batchsize 100 | batchsize 200 | batchsize 300 |
> | ----------------- | :-----------: | :-----------: | ------------- |
> | CE  + Regularizer |     39.08     |     40.07     | 40.14         |

---

### Official Review · Reviewer_DbNo · 2022-10-23

**Confidence:** 4
**Correctness:** 3
**Technical Novelty And Significance:** 2
**Empirical Novelty And Significance:** 3
**Recommendation:** 6

**Clarity, Quality, Novelty And Reproducibility:**

In general, the paper is not easy to follow, especially the motivation for the proposed method. I'm fine with each component of the proposed method is not new, such as self-supervised head, l1 and l2 regularization, but the motivation of combining them together is important to me.

**Strength And Weaknesses:**

Strength:
The proposed framework is extendable and can be plugged into other robust loss functions to further improve performance.

Extensive experiments validate the effectiveness of the proposed method, and theoretical analyses are provided to support the claim.

Weakness:
The motivation of the method is hard to follow and not convincing. Although the authors hope to use representation regularizers to cut off some redundant function space without hurting the optima, the actual method proposed in the paper seems to be not related to what the author claims.  Also, what’s the high-level motivation of the proposed method? How does the self-supervised branch work in the method?

In table 2 and table 3, the improvement of the proposed regularization loss is limited compared with previous methods.

Are there any large real-world datasets with label noise other than toy datasets, such as CIAFR variants? If yes, what’s the empirical performance of those datasets?



**Summary Of The Paper:**

This paper proposes a new approach to tackle the problem of learning with noisy labels from the perspective of representation regularization. In detail, the self-supervised learning (SSL) head and supervised noisy labels head (SL) after the feature extractor (encoder) are utilized to extract SSL features and SL features respectively, and then a regularization loss is proposed to minimize the distance between the SSL features and SL features.

**Summary Of The Review:**

With the findings above, I currently give the paper a reject score.

Update: I raised my score to 6 after discussing with AC and other reviewers.

---

> ### Author Response · Authors · 2022-11-13
> **Response to Reviewer DbNo**
>
> Thanks for your detailed comments and suggestions. Our responses for your questions are as follows:
>
> **Question 1:** The motivation for the proposed method.
>
> **Response 1:** We respectfully do not think that our proposed method is a simple combination of existing approaches. The motivation of using (self-supervised) representation to mitigate memorization of learning with noisy labels is inspired by our analysis on the generalization error of DNN (Section 3).
>
> Theorem 1 and Corollary 1 in Section 3, along with the the supportive experiments in Figure 4 and Figure 5 give us insights that learning with an unfixed encoder is not stable, which may overfit noisy patterns and converge to a poor local optimum. Restricting the search space makes the convergence stable (reducing estimation error) with the cost of increasing approximation errors. Thus this motivates us to use regularization to compromise between a fixed and unfixed encoder.
>
> **Question 2:** How does the self-supervised branch work in the method?
>
> **Response 2:** The representation coming from self-supervised learning (SSL) is unbiased since SSL training does not involve labels. This is important for designing our proposed regularizer by utilizing SSL features to alleviate over-fitting effect ($L_{c}$ in Equation (4)). One can think of that Equation (4) is using SSL features to **correct** SL features.
>
> Ideally, if the SSL features from different classes are well separated (large $||u_1 - u_2||$ and low $\Sigma$ in Assumption 3), then Theorem 2 implies that the DNN has very low generalization error or overfitting effect.
>
> **Question 3:** In table 2 and table 3, the improvement of the proposed regularization loss is limited compared with previous methods.
>
> **Response 3:** First of all, achieving SOTA is not our main purpose. Our main goal is to show that our regularizer can be easily incorporated into other robust losses or methods.
> Table 1 has demonstrated its efficiency. For example, when symmetric label noise ratio is 0.8. The regularizer can improve existing approaches over 20\% points which we assume is a significant improvement.
>
> It is also worth noting that in Table 2 and Table 3, we only use CE as our base loss functions to combine our regularizer. The method of CE + Regularizer is already competietive to popular approaches. For example, CE + Regularizer outperforms other popular approaches over 5 points for CIFAR10N in Table 2.
>
> **Question 4:** Are there any large real-world datasets with label noise other than toy datasets, such as CIAFR variants?
>
> **Response 4:** CIFAR10N and CIFAR100N [R1] are the variants of CIFAR which contain real-world label noise making it very appropriate for evaluating methods. We evaluate our regularizer for CIFAR10N and CIFAR100N in Table 2.
>
> Clothing1m [R2] is a large real-world dataset that contains real-world label noise. We evaluate our regularizer for Clothing1m in Table 3.
>
>
> [R1] Learning with Noisy Labels Revisited: A Study Using Real-World Human Annotations. ICLR 2021
>
> [R2] Learning From Massive Noisy Labeled Data for Image Classification. CVPR 2015

---

### Official Review · Reviewer_L66U · 2022-10-23

**Confidence:** 4
**Correctness:** 2
**Technical Novelty And Significance:** 3
**Empirical Novelty And Significance:** 3
**Recommendation:** 3

**Clarity, Quality, Novelty And Reproducibility:**

- Clarity
    - the paper contains a great amount of information, a little bit hard to fully appreciate all the contributions.
    - how do Theorem 1 and Corollary 1 contribute to the remaining discussion of this paper other than offering some generic discussion in the insight section?
    - observations 1-4 are more like hypotheses

- Quality
    - Theorem 1 seems to have some issues
        - no definition for Ne, or maybe e.
        - in the proof body of Theorem 1, the term Ne also suddenly appears with no noticeable context.
    - Theorem 2 also has issues in its body
        - why the body needs to specific an extra term N if N is infinity, and batchsize B, since neither of them appear in the theorm body
        - no clear definition of \Sigma, the only one I can find is in assumption 3, but both usages appear generic, thus no clear evidence saying they are referring to the same thing.
        - Theorem 2 needs assumptions 1-3 should to be specified in its own body, not as free texts above.

- Novelty:
   - learning with regularization over representations has been discussed elsewhere in the broader scope of noise and robust learning e.g., [1], but is probably novel in this setting, thus it should be fine.
        - [1] Toward Learning Robust and Invariant Representations with Alignment Regularization and Data Augmentation

**Strength And Weaknesses:**

- strength
    - the paper studies an interesting problem on learning with noise labels
    - the paper is condensed with theoretical discussions, might offer interesting results in a broader scope
- weakness
    - the discussion of the theoretical result does not seem rigorous enough (see below)
    - the paper contains a lot of contents, some contents might be better to be moved to appendix. It's quite hard to parse all the theoretical information within the main manuscript.


**Summary Of The Paper:**

This paper tries to use regularization between learned representations to study the problem of learning with noise features. The paper offers some theoretical discussions, and proposes a new methods, however, I do not see how the theory is connected with the proposed method other than a generic lesson learned on "restricting the search space with regularization is a reasonable approach", which is true, but probably we do not need an entire theoretical discussion for this lesson.

**Summary Of The Review:**

It's a nice and interesting work, but there are too many issues in the theoretical discussions, I think the work needs a deep and rigorous pass on the theoretical discussions before being published.

---

> ### Author Response · Authors · 2022-11-13
> **Response to Reviewer L66U**
>
> Thanks for your detailed comments and suggestions. Our responses for your questions are as follows:
>
> **Question 1:** Concerns regarding Clarity.
>
> **Response 1:**
>
> **Implication of Theorem 1 and Corollary 1:** Theorem 1 and Corollary 1 do not relate to our proposed regularizer in Section 4 while Theorem 2 does. Theorem 1 and Corollary 1 are mainly to show how model capacity affect network performance for different noise type. Theorem 1, Corollary 1 and the supportive experiments in Figure 4 and Figure 5 give us insights that learning with an unfixed encoder is not stable, which may overfit noisy patterns and converge to a poor local optimum. Restricting the search space makes the convergence stable (reducing estimation error) with the cost of increasing approximation errors. Thus this motivates us to use regularization to compromise between a fixed and unfixed encoder.
>
> **Question 2:** Concerns regarding Quality.
>
> **Response 2:**
>
> **meaning of $Ne$:** Sorry for the confusion. $Ne$ stands for $N\cdot e$ where $N$ is the number of samples and $e = 2.718$ is the base of the natural logarithms.
>
> **infinity of $N$:** Since we prove the theorem using the loss expectation (Theorem 2). The expectation itself implies that we have infinite samples. Using expectation to prove loss robustness is common in the literature of learning with noisy labels. For example, [R1, R2, R3]. We will make it more clear in the revised version.
>
> [R1]: Robust Loss Functions under Label Noise for Deep Neural Networks AAAI 2017
>
> [R2]: Normalized Loss Functions for Deep Learning with Noisy Labels. ICML 2020
>
> [R3]: Peer Loss Functions: Learning from Noisy Labels without Knowing Noise Rates. ICML 2020
>
> **no clear definition of $\Sigma$:** $\Sigma$ in Assumption 3 and Theorem 2 are referring the same thing. $\Sigma$ denotes the covariance matrix of the distribution of $h(f(X_{+}))$ and $h(f(X_{-}))$ in Assumption 3. Intuitively, if $tr(\Sigma)$ is big, then the representations of $h(f(X_{+}))$ and $h(f(X_{-}))$ are hard to separate, resulting in lower performance which is revealed in Theorem 2.
>
> **position of assumptions:** Thanks for your suggestion. We have revised accordingly.
>
> **Question 3:** Concerns regarding Novelty.
>
> **Response 3:** The paper [R4] proposes the regularization based on the alignment rather than self-supervised representation and does not address the problem of learning with noisy labels explicitly. However, we thank the reviewer for bringing [R4] to our attention. We will cite and discuss [R4] in the revised version.
>
> [R4] Toward Learning Robust and Invariant Representations with Alignment Regularization and Data Augmentation. KDD 2022

---

### Official Review · Reviewer_kNp2 · 2022-10-25

**Confidence:** 4
**Correctness:** 3
**Technical Novelty And Significance:** 3
**Empirical Novelty And Significance:** 2
**Recommendation:** 8

**Clarity, Quality, Novelty And Reproducibility:**

This paper is well-written and states its contributions clearly. As far as I am aware, the idea to fix a model's encoder as a way to regularize its representations is novel.

The key improvement that this paper needs in its writing is in regards to the use of citep vs cite. There are several repeated instances where citep should've been used instead of cite e.g. line 9 of second paragraph of introduction

**Strength And Weaknesses:**

### Strengths
- The key insight of this work is that constraining the representation learning portion of a classifier, to be fixed, helps regularize the ability of the classifier to memorize noisy labels. This insight is demonstrated effectively through the theorems and empirical analyses.


### Weaknesses

- While the theorems are helpful, generally, it is a bit hard for me to read too much into whether they should be meaningful. In the sense that we now know that the VC dimension term, for neural networks, might not encode as much information as we thought about the generalization behavior of the neural networks hypothesis classes.

- Can the authors comment on how other performance properties of these trained models are affected by noise in labels? For example, should these insights also translate to something like the model's calibration?

- The insights from section 5.1 are interesting and it might make sense to discuss these in the introduction. However, the results seem to suggest that one needs to one which noise regime that one is in. For example, for a task, in practice, I would have to know when the data has high symmetric label noise or how much bias to expect. These seem like challenging things to know ahead of time.

- Some papers that might be useful to consider: Feldman et. al. (Does learning require memorization? a short tale about a long tail), and Li et. al. (How does a Neural Network's Architecture Impact its Robustness to Noisy Labels?)

**Summary Of The Paper:**

This paper analyzes how the representation learned by a DNN and its downstream generalization is affected by noise in the training labels. The key insight in this paper is that the authors suggest to decouple the classifier into an encoder and a linear model. The encoder is fixed for a downstream task, while the 'linear' model component is fine-tuned. This insight comes from the fact that 'smaller' (in terms of VC dimension) hypothesis classes have more difficulty learning label noise. Consequently this can serve as a regularizer on the task. Given noisy label scenario, the paper gives an upper bound on the estimation error for a particular hypothesis class that depends on the amount of label noise (through a $(1 - \epsilon^2)$ term where $\epsilon$ is the noise parameter), and VC dimension of the model class. A version of this theorem for a fixed decoupled encoder+linear model setting  suggests that the fixed encoder setting leads to a reduction in the ability of the class to fit noisy labels at the expense of higher approximation errors. Overall, this  insight is instantiated on an SSL setting where a precise relationship is revealed between the quality of the representations and the error on noisy examples. The paper concludes with experiments to back up this claim.

**Summary Of The Review:**

This paper is well-written and clear. Overall, the paper provides an important insight about how to stem a model's reliance on  noisy labels using a fixed encoder and regularizing a model's representations.

---

> ### Author Response · Authors · 2022-11-13
> **Response to Reviewer kNp2 (Part 1)**
>
> Thanks for your detailed comments and suggestions. Our responses for your questions are as follows:
>
>
> **Question 1:** While the theorems are helpful, generally, it is a bit hard for me to read too much into whether they should be meaningful. In the sense that we now know that the VC dimension term, for neural networks, might not encode as much information as we thought about the generalization behavior of the neural networks hypothesis classes.
>
> **Response 1:** Thanks for the comments. The VC dimension term is only for characterizing the model capacity. The main factors that we want to highlight, for example, in Theorem 1, is the order wrt $N,|\mathcal C|$, and $\varepsilon$. Specifically, the takeaway message from Theorem 1 is the error is in the order of $O(\sqrt{\frac{|\mathcal C| \log(N/|\mathcal C|)}{N(1-\varepsilon)^2})}$, and it could be more concise if we ignore the less-dominate logarithmic terms, i.e., $O(\sqrt{\frac{|\mathcal C|}{N(1-\varepsilon)^2})}$. We are aware that a tight generalization bound for an arbitrary neural network is complicated and an open question. We did not get into it and just use a traditional term to summarize the possible effect of networks with different capacity.
>
> we acknowledge that building on more informative generalization results for neural network would indeed be more relevant to our observations in practice but the classical ones we used gives us a concise way to make the comparisons intuitive and straightforward. The experiments in Figure 4, Figure 5 and Table 5 in the paper also confirm our analyses.
>
> **Question 2:** Can the authors comment on how other performance properties of these trained models are affected by noise in labels? For example, should these insights also translate to something like the model's calibration?
>
> **Response 2:** If the model calibration refers to the probabilities predicted by a model, the label noise will make the model mis-calibrated. For example, consider a binary classification between cats and dogs. If one image in the training dataset has 10 labels, where 6 of them are CAT (true label) and 4 of them are DOG (wrong label), the model's prediction probability on CAT would be 0.6 and on DOG would be 0.4 if the cross-entropy loss is minimized on this image. One approach to correct this model mis-calibration is loss correction [R1--R3]. The idea is use a transition matrix to invert (correct) the mis-calibrated probability vector $[0.6,0.4]$ back to $[1,0]$. Our proposed regularizer can also correct the mis-calibrated probability since the high-level intuition of the regularizer ($L_{c}$ in Equation 4) is to use unbiased self-supervised features to correct the supervised features (mis-calibrated probability).
>
>
> [R1] Learning with noisy
> labels. NeurlPS 2013.
>
> [R2] Classification with noisy labels by importance reweighting. TPAMI 2015
>
> [R3] Making deep
> neural networks robust to label noise: A loss correction approach. CVPR 2017

---

> ### Author Response · Authors · 2022-11-13
> **Response to Reviewer kNp2 (Part 2)**
>
> **Question 3:** The insights from section 5.1 are interesting and it might make sense to discuss these in the introduction. However, the results seem to suggest that one needs to one which noise regime that one is in. For example, for a task, in practice, I would have to know when the data has high symmetric label noise or how much bias to expect. These seem like challenging things to know ahead of time.
>
> **Response 3:**
>
> $\bullet$ There are some works in the literature of learning with noisy labels [R4,R5] that assume the noise rate is known before applying the learning methods since in practice, one can sample a subset from the whole data or using the validation set to roughly estimate the noise ratio [R4, R6, R7], or use algorithms to estimate the noise ratio [R8]. After estimating the ratio, we can decide which method to apply for learning a robust classifier.
>
> $\bullet$ We can also decide whether to fix the encoder by examining the data distribution without estimating the noise ratio. Corollary 1 implies that if the encoder is learned very well, i.e.,
> $\textsf{Error}\_{A}(C\_{\mathcal D}^{\mathcal G \circ \mathcal F},C^*) \approx  \textsf{Error}\_{A}(C\_{\mathcal D}^{\mathcal G | f},C^*)$
> fixing the encoder has benefits over unfixed encoder even when noise rate is small. Figure 5 further supports this. Since for DogCat, CIFAR10 and CIFAR100 dataset, each class have 12500 samples, 5000 samples and 500 samples, respectively. When applying the self-supervised learning on these datasets, the encoder quality is $\text{DogCat} > \text{CIFAR10} > \text{CIFAR100}$. From Figure 5 (a)(b)(c), the crossing points between fixed and unfixed encoders are 0.0, 0.4,0.7.  Thus if a dataset has more samples per class, we can choose to use fixed encoder for learning a robust classifier.
>
> [R4] Co-teaching: Robust Training of Deep Neural Networks with Extremely Noisy Labels. NeurlPS 2018
>
> [R5] How does disagreement help generalization against label corruption? ICML 2019
>
> [R6] Classification with noisy labels by importance reweighting. TPAMI
>
> [R7] An efficient and provable approach for mixture proportion estimation using linear independence assumption. CVPR 2018
>
> [R8] Clusterability as an alternative to anchor points when learning with noisy labels. ICML 2021.
>
> **Question 4:** Some papers that might be useful to consider: Feldman et. al. (Does learning require memorization? a short tale about a long tail), and Li et. al. (How does a Neural Network's Architecture Impact its Robustness to Noisy Labels?
>
> **Response 4:** Thanks for bring these papers to our attention. The first paper [R9] states the memorization is necessary even when dataset has noisy labels. However, the setting in the first paper considers that the noise level is relatively low and affects primarily hard examples. Thus it does not contradict with the commonly beliefs in the literature of learning with noisy labels that over memorization is bad for performance and requires special treatment [R10].
>
> The second paper [R11] analyses how DNN architectures affect the robustness to noisy labels from the alignment perspective, i.e., a network is more robust to noisy labels if its architecture is more aligned with the target function than the noise. While we analyze the robustness of DNN structure to noisy labels from the perspective of estimation error and approximation error. Interestingly, some experiments in [R11] also supports our analyses. For example, Table 1 in [R11] also shows that a shallow network performs better than deeper network under higher uniform (symmetric) label noise for vanilla (CE) training.
>
> We will add these analyses in the revised version.
>
> [R9] Does Learning Require Memorization? A Short Tale about a Long Tail. STOC 2020
>
> [R10] Understanding Instance-Level Label Noise: Disparate Impacts and Treatments. ICML 2021.
>
> [R11] How Does a Neural Network’s Architecture Impact Its Robustness to Noisy Labels? NeurlPS 2021
>
> **Question 5:** Inconsistency of cite and citep.
>
> **Response 5:** Thanks for pointing out. We have fixed this issue.

---

### Official Review · Reviewer_Y66C · 2022-10-30

**Confidence:** 3
**Clarity, Quality, Novelty And Reproducibility:** 1. The use of contrastive learning fo…
**Correctness:** 4
**Technical Novelty And Significance:** 3
**Empirical Novelty And Significance:** 2
**Recommendation:** 8

**Strength And Weaknesses:**

## Strengths
1. The key strength of this work is that it utilizes the advances in contrastive learning to regularize the representations in a supervised learning framework.
2. The experimental observations complement some of the theoretical implications. (I was not able to appreciate the theoretical results and found them less intuitive, but I would like to see what other reviewers feel about it)
3. The regularization method can be added on top of any existing method for training under label noise and results in consistent gains on the CIFAR10 dataset.
4. I would have liked to see a larger portion of the paper devoted to the empirical analysis and ablations which is the strength of the work in my view.


## Weaknesses and Questions:
1. Theoretical Analysis: The theoretical analysis and assumptions seem to be very distant from the empirical approach.
   (1) The notations are very confusing and non-succinct.
   (2) The assumptions such as 0 variance, and infinite training data size are unrealistic.
   (3) Connection between the theoretical results and the empirical approach is weak -- Gaussian features and discussion on the relevance of the theoretical analysis on the empirical observations.

2. Figure 2 is very confusing till we read the first paragraph of page 5. Similarly, in Figure 1, what is the training setup? these things should ideally be self-contained in the captions.

3. What is the problem setup in Figures 4 and 5? What is the noise level for asymmetric and instance-level noise? How many samples get down-sampled?

4. Experimental Configurations: How was the value of $\lambda$ selected? What was the pertaining dataset for the encoder? Which augmentations were used?

4. Comparisons with state of art defenses such as ELR and SOP are missing. Refer to this paper for a list of more competitive baselines: https://proceedings.mlr.press/v162/liu22w.html

5. It is suggested to fix the encoder for high symmetric label noise based on Figure 5. Are you doing this for the table numbers?

6. How is observation 3 explained? What is its relevance? Can you provide some more information about the same?

7. Results in Table 1 are only provided for the CIFAR10 dataset. This is not comprehensive enough.

## Other Comments
1. \citet{} and \citep{} have been used incorrectly throughout the paper. Please fix this.
2. Figure 4 appears after Figure 5



**Summary Of The Paper:**

This work discusses how regularizing the representations of deep neural networks can help improve model generalization when the dataset is noisy. In particular, the authors first discuss how limiting the capacity of a neural network can help improve generalization in noisy data settings. This is achieved via the regularization of the representation space of the model. Regularization is performed by penalizing the distance between the representations from the projection head of SSL and SL features. Experimental results are demonstrated on various noise types and noise severities to suggest that the additional regularization objective can help improve the generalization performance of various noisy data training algorithms.

**Summary Of The Review:**

The paper shows strong empirical gains in mislabeled data settings, but misses comparisons with state-of-the-art methods. I am unsure about the significance of the theoretical results.

Raising score to 8 after author rebuttal

---

> ### Author Response · Authors · 2022-11-13
> **Response to Reviewer Y66C (Part 1)**
>
> Thanks for your detailed comments and suggestions. Our responses for your questions are as follows:
>
> **Question 1**:Theoretical analysis: assumptions; connection between the theoretical results and the empirical approach.
>
> **Response 1:**
>
> **Zero-variance assumption:** Theorem 2, which uses the zero variance assumption, shows the optimal (convergence) state when applying the regularizer.  The zero variance is validated in Table 6 (Appendix D.5, page 23). It shows that the variance of noisy samples in each label id is close to 0 when network converges.
>
> **Infinite data size:** Sorry for the confusion. We prove the theorem using the loss expectation (Theorem 2). Please note the expectation itself implies that we have infinite samples. Using expectation to prove loss robustness is common in the literature of learning with noisy labels. For example, [R1, R2, R3]. We have made it clear in the revised version.
>
> [R1]: Robust Loss Functions under Label Noise for Deep Neural Networks AAAI 2017
>
> [R2]: Normalized Loss Functions for Deep Learning with Noisy Labels. ICML 2020
>
> [R3]: Peer Loss Functions: Learning from Noisy Labels without Knowing Noise Rates. ICML 2020
>
> **Connection between theorems and empirical approach** We use Gaussian assumption for self-supervised features mainly for finding a closed-form solution in Theorem 2. Besides, some literatures also observe that self-supervised features tend to follow Gaussian distribution [R4].
>
> Theorem 2 is to show the effectiveness of the regularizer, I.e., $L_{c}$ in Equation (4), which implies that using the regularizer, we can prevent network from over-fitting to noisy labels with well-learned self-supervised features. Figure 6 and Table 1 supports the Theorem 2 by observing the network performance of early and latter training stages (Best and last epoch accuracy).
>
> [R4]: Understanding contrastive representation learning through alignment and uniformity on the hypersphere. ICML 2020
>
> **Question 2**: Figure 2 is very confusing till we read the first paragraph of page 5. Similarly, in Figure 1, what is the training setup? these things should ideally be self-contained in the captions.
>
> **Response 2:** Sorry for putting Figure 2 in early page while the pointer is actually in page 5. We have revised this part. The training setup is the same as the baseline setup in Table 1 in the paper.
> Please note the common experimental settings are included in Appendix E on page 23. We are sorry for not presenting details in the main paper due to the space limit. We have added the setup in the caption and elaborated more details in the supplementary material (Appendix E, page 23).
>
> **Question 3**: What is the problem setup in Figures 4 and 5? What is the noise level for asymmetric and instance-level noise? How many samples get down-sampled?
>
> **Response 3:** Please note the common experimental settings are included in Appendix E on page 23. Sorry again for not presenting details in the main paper due to the space limit.
>
> Figure 4 is examining the performance of fixed v.s. unfixed encoder on symmetric label noise. As shown in the figure, the x-axis denotes noise rate (asymmetric: [0.1, 0.2, 0.3,0.4], instance-dependent: [0.2, 0.3, 0.4, 0.5, 0.6]).
>
> Figure 5 is examining the performance of fixed v.s. unfixed encoder on asymmetric and instance-dependent label noise. Please note the noise rates have also been presented by the x-axis. The number of samples which get down-sampled varies for different noise rates and different classes since for instance dependent label noise, the number of samples for each class is different. The guideline is to  $\mathbb P(\widetilde{Y}=i) = \mathbb P(\widetilde{Y}=i)$. For example, for noise rate 0.6, we find that the number of overall down-sampled samples for all the classes is $21\\% \cdot N$, where $N$ is the number of samples in the dataset.
>
> **Question 4**: Experimental Configurations: How was the value of $\lambda$ selected? What was the pertaining dataset for the encoder? Which augmentations were used?
>
> **Response 4**: $\lambda$ is a hyper-parameter. We keep $\lambda = 1$ for all the experiments in Table 1, as noted in the caption of Table 1.
>
> The pre-training dataset is the same dataset as we use for learning with noisy label tasks which means that we do not include external dataset for training. We included the detailed setup of pre-training in Appendix E of the original submission. The augmentation is consistent with [R5] that includes random resize and crop, random horizontal Flip, etc. The details can also be found in our realeased code.
>
> [R5] A simple framework for contrastive learning of visual representations. ICML 2020

---

> ### Author Response · Authors · 2022-11-13
> **Response to Reviewer Y66C (Part 2)**
>
> **Question 5:** Comparisons with state of art defenses such as ELR and SOP are missing.
>
> **Response 5:** Achieving SOTA is not our main goal. Our main purpose is to show that our proposed regularizer can further improve performance by incorporating our regularizer with popular robust losses such as GCE, Peer loss in Table 1. For example, in Table 1, when symmetric label noise ratio is 0.8. The regularizer can improve existing approaches over 20\% points which we think is a big improvement.  We **did compare with ELR in Table 2 and Table 3** on the datasets with real-world label noise.
>
> **Question 6:** It is suggested to fix the encoder for high symmetric label noise based on Figure 5. Are you doing this for the table numbers?
>
> **Response 6:** No. Table 1 is to show that our proposed regularizer can further improve performance by incorporating our regularizer with popular existing approaches. Since the compared approaches in Table 1 has no pre-training step, for a fair comparison, we do not perform pre-training and fixing encoder. I.e., all the methods in Table 1 are trained from scratch.
>
> **Question 7:** How is observation 3 explained? What is its relevance? Can you provide some more information about the same?
>
> **Response 7:** Observation 3 suggests that "Do not fix encoder when bias exists". This observation is supported by Figure 4 and can be explained by Theorem 1. Since the asymmetric and the instance-dependent label noise do not satisfy noise consistency (Definition 1), fixing the encoder will induce large estimation error for such noise. Figure 4 shows the unfixed encoder is generally better than the fixed encoder for both asymmetric and instance-dependent noise, from which we summarize this observation.
>
> **Question 8:** Results in Table 1 are only provided for the CIFAR10 dataset. This is not comprehensive enough.
>
> **Response 8:** We provide experiments on CIFAR10-N, CIFAR100-N and Clothing1m in Table 2 and Table 3 to verify the effectiveness of the regularizer. We have also conducted experiments on CIFAR-100 (hyper-parameters are the same as CIFAR10 in Table 1). The results are as follows:
>
> | Method         | symmetric noise ratio 0.8 | symmetric noise ratio 0.5 |
> | -------------- | :-----------------------: | :-----------------------: |
> | CE             |           20.53           |           15.67           |
> | CE+Regularizer |           28.19           |           22.95           |
>
> We will conduct more experiments and include them in the revised version.
>
>
> **Question 9:** Inconsistency of cite and citep
>
> **Response 9:** Thanks for pointing out. We have fixed this issue.

---

> > ### Comment · Reviewer_Y66C · 2022-11-18
> > **Regarding Comparisons with baselines**
> >
> > The author's response to Question 5 is discomforting. It is unethical to report baselines when the performance of a proposed method is better than them and conveniently ignore them in the tasks that the approach is worse off on.

---

> > > ### Author Response · Authors · 2022-11-19
> > > **Response to the comparisons with baselines**
> > >
> > > We think there may exist some misunderstandings. We report ELR in Table 2 and Table 3 and do not report ELR in Table 1 because the purpose of Table 1 is to show our
> > > proposed regularizer can incorporate popular robust losses to further improve performance. Since ELR itself is a regularization method, i.e., the loss of ELR is a base loss (CE)
> > > plus regularization, comparing ELR in Table 1 will result in base loss plus two regularizations. Thus it may not be clear which regularization that the improvement comes from.
> > > Nevertheless, we have used the official implementation from ELR loss (https://github.com/shengliu66/ELR/blob/master/ELR/model/loss.py) to conduct the experiment on CIFAR10. The hyper-parameter settings such as learning rate, batch size, optimizer, etc, are consistent with all the methods in Table 1 except that we change the base loss to ELR. The results are as follows:
> > >
> > > |                        | symmetric noise ratio 0.6 | symmetric noise ratio 0.8 | aymmetric noise ratio 0.4 |
> > > | :--------------------: | :-----------------------: | :-----------------------: | :-----------------------: |
> > > |          ELR           |        72.56/71.58        |        42.67/23.57        |        86.65/85.27        |
> > > | ELR +  our Regularizer |      **76.95/75.98**      |      **63.4/57.45**       |      **88.62/87.92**      |
> > >
> > >
> > > It can be observed that ELR with our regularization also outperforms ELR. These results have been merged to Table 1 in the main paper. We have also attached the codes of all the methods in Table 1 in the supplementary material.

---

> > > > ### Comment · Reviewer_Y66C · 2022-11-29
> > > > **Follow Up**
> > > >
> > > > Thank you for the response and clarification with these numbers. These look strong and support the efficacy of the method.
> > > >
> > > > With regards to results on CIFAR100, can you please provide more comprehensive results as is done in state-of-art works. (example Table 1 here https://arxiv.org/pdf/2007.00151.pdf)
> > > >
> > > > It is confusing why the numbers are not in the same ball-park, plus I would like to see if regularization helps on top of SOTA methods even for CIFAR-100.
> > > >
> > > > Thanks!

---

> > > > > ### Author Response · Authors · 2022-12-02
> > > > > **Further Response**
> > > > >
> > > > > Very thanks for the reply.
> > > > >
> > > > > - We report two numbers for each setting, i.e., best epoch test accuracy and last epoch test accuracy, presented as best/last. Some methods may achieve a higher best epoch accuracy but very low last epoch accuracy due to the memorization of noisy labels. It may cause harm to the system if we deploy such methods on the online classfication system which is required to continually learn from the data. In this paper, we want to show that our proposed regularizer can further improve robust losses in terms of both best epoch accuracy and last epoch accuracy.
> > > > >
> > > > > - Following your suggestions, we have implemented all the SOTA methods in Table 1 of https://arxiv.org/pdf/2007.00151.pdf including Bootstrap [R1], Forward [R2], GCE [R3], SL [R4] and ELR [R5], and tested our regularizer with these SOTA methods on CIFAR100. Specifically, for Forward and GCE, we use the same implementation as we perform on CIFAR10; For Bootstrap, we use the implementation from https://github.com/vfdev-5/BootstrappingLoss/blob/master/bootstrapping_loss/loss.py and set the $\beta$ hyper-parameter in Bootstrap  loss to be 0.8 as the paper suggests; For SL, we use the impelmentation from https://github.com/HanxunH/SCELoss-Reproduce and set the $\alpha$ and $\beta$ hyper-parameter in SL loss to be 6.0 and 1.0 as the paper suggests; For ELR loss, we use the official implementation from https://github.com/shengliu66/ELR/blob/master/ELR/model/loss.py. The other training hyper-paramter for CIFAR100 including learning rate, number of training epochs, optimizer, etc, are the same for all the methods. The results are as follows:
> > > > >
> > > > > |                             | symmetric noise 0.6 | symmetric noise 0.8 | asymmetric noise  0.4 |
> > > > > | :-------------------------: | :-----------------: | :-----------------: | :-------------------: |
> > > > > |             ELR             |     34.36/14.21     |     18.95/5.54      |      55.36/33.5       |
> > > > > |    ELR + our Regularizer    |   **37.17/23.45**   |   **20.92/13.16**   |    **56.55/36.49**    |
> > > > > |          Bootstrap          |     34.21/13.8      |     18.45/6.68      |      39.28/30.93      |
> > > > > | Bootstrap + our Regularizer |   **35.63/27.88**   |   **19.92/16.3**    |    **40.21/34.56**    |
> > > > > |           Forward           |     38.72/22.26     |     17.29/8.15      |    42.35/**40.46**    |
> > > > > |  Forward + our Regularizer  |   **39.01/29.39**   |   **26.92/21.02**   |    **43.35**/40.02    |
> > > > > |             SL              |     34.16/14.12     |   19.17/**6.68**    |      40.76/32.07      |
> > > > > |    SL + our Regularizer     |   **34.20/15.3**    |   **19.65**/6.02    |    **40.81/34.05**    |
> > > > > |             GCE             |     38.08/26.79     |     23.24/16.9      |      41.26/31.4       |
> > > > > |    GCE + our Regularizer    |   **44.27/43.48**   |   **35.01/34.55**   |    **42.35/35.74**    |
> > > > >
> > > > >
> > > > > We find that the best epoch accuracies for all the SOTA methods do not vary too much for symmetric label noise on CIFAR100. However, ELR is very superior for asymmetric label noise.  Our proposed Regularizer can improve these SOTA methods in terms of best epoch accuracy and last epoch accuracy in most of cases. Especially, we find that our proposed regularizer works very better when we combine our regularizer with GCE loss.
> > > > >
> > > > > Best.
> > > > >
> > > > > [R1] Training deep neural networks on noisy labels with bootstrapping
> > > > >
> > > > > [R2] Making deep neural networks robust to label noise: A loss correction approach.
> > > > >
> > > > > [R3] Generalized cross entropy loss for training deep neural networks with noisy labels
> > > > >
> > > > > [R4] Symmetric cross entropy for robust learning with noisy labels.
> > > > >
> > > > > [R5] Early-Learning Regularization Prevents Memorization of Noisy Labels

---

> > > > > > ### Comment · Reviewer_Y66C · 2022-12-05
> > > > > > **Follow Up**
> > > > > >
> > > > > > Thank you for your efforts! The gains look consistent and impressive. I have one more confusion. Can you please clarify why you report numbers in the extremely high noise regime, and not the low noise regime like 20,40% for symmetric and 10-20 for aymmetric as well.
> > > > > >
> > > > > > Thanks!

---

> > > > > > > ### Author Response · Authors · 2022-12-06
> > > > > > > **Further Response**
> > > > > > >
> > > > > > > Thank you for your reply.
> > > > > > >
> > > > > > > - As we stated in our previous response, some methods may achieve a higher best epoch accuracy but very low last epoch accuracy due to the memorization of noisy labels. It happens when the noise rate is high. When noise rate is relatively low, the gap between the best epoch accuracy and last epoch accuracy becomes small. Since the motivation of our regularizer is to mitigate the memorization effect, we test our regularizer in high noise regime to show the benefits of our regularizer.
> > > > > > >
> > > > > > > - We additionally perform experiments for low noise regime as follows. The experiment setting is the same as we did for high noise regime in our last response.
> > > > > > >
> > > > > > > |                             | symmetric noise ratio 0.2 | asymmetric noise ratio 0.2 |
> > > > > > > | :-------------------------: | :-----------------------: | :------------------------: |
> > > > > > > |             ELR             |        53.64/39.8         |        58.63/47.96         |
> > > > > > > |    ELR + our Regularizer    |      **57.26/44.47**      |      **61.28/48.46**       |
> > > > > > > |          Bootstrap          |        50.77/38.71        |         49.12/44.8         |
> > > > > > > | Bootstrap + our Regularizer |      **50.81/46.7**       |      **52.13/48.31**       |
> > > > > > > |           Forward           |      56.99/**53.92**      |      57.63/**55.87**       |
> > > > > > > |  Forward + our Regularizer  |      **57.17**/52.75      |      **57.89**/54.17       |
> > > > > > > |             SL              |        50.21/35.04        |        50.01/42.84         |
> > > > > > > |      SL + Regularizer       |      **50.87/42.03**      |      **52.44/48.59**       |
> > > > > > > |             GCE             |        52.23/42.46        |        53.86/43.18         |
> > > > > > > |      GCE + Regularizer      |      **55.79/55.5**       |      **57.26/50.83**       |
> > > > > > >
> > > > > > > It can be observed that our regularizer still improves these SOTA methods in most of cases and our regularizer works better when we combine our regularizer with GCE loss. The improvements are consistent and obvious but not as very significant as in the high noise regime since the SOTA methods are less affected by memorization effect in low noise regime. We will add these results and analyses in the revised version.
> > > > > > >
> > > > > > > Best.

---

> > > > > > > > ### Comment · Reviewer_Y66C · 2022-12-06
> > > > > > > > **Thanks**
> > > > > > > >
> > > > > > > > Thank you for the clarification. I have raised my score to 8.

---

### Author Response · Authors · 2022-11-14
**Response to all the reviewers**

Dear reviewers,

Thanks for your constructive comments and suggestions which help us improve our paper. We have responded to your concerns in the rebuttal and uploaded the revision. In particular,

- We explained the motivation of our paper and the implication of the Theorems.

- Some notations and terms are revised and re-defined for clarity.

- More experiments regarding to the effect of batch-size, disentangled representaitons, etc, are performed.

- We followed most of the other suggestions and have revised the paper accordingly.

We have highlighted the major changes in color blue.

Thank you and we look forward to continuing the discussion.

Best,

Authors

---

### Decision · Program_Chairs · 2023-01-20

**Decision:**

Accept: poster

**Justification For Why Not Higher Score:**

Theoretical results appear confusing in places.

**Justification For Why Not Lower Score:**

Good empirical results, as noted by majority of reviewers. Some theoretical analysis.

**Metareview: Summary, Strengths And Weaknesses:**

The paper proposes a new regularizer for training neural networks under noise. The idea is to decompose such a network into an encoder and classifier. The encoder is regularized based on a semi-supervised objective, so that supervised and self-supervised representations align. There is some theoretical analysis that this can lead to better generalization. This is followed by empirical analysis showing the benefits of this approach.

The original reviews were a bit mixed, with critiques focussing on some missing baselines, and lack of results on CIFAR100 and other more challenging datasets. Through the response which provided new results, most reviewers were satisfied, and had a positive impression of the paper. My own assessment is more reserved, based on some unclear parts in the theoretical analysis (which constitutes the bulk of the paper). The results make some abstract sense --- essentially, training an encoder with sufficient regularization can prevent overfitting to noise --- but the current presentation is a bit confusing in parts. There is also a lack of clarity how these results compare to existing works in the literature. I will uphold the final reviewer consensus, given that the empirical results are indeed very promising, but strongly advise the authors to thoroughly clarify the theoretical section.

Some specific comments:
- Aren't Lemmas 1, 2 well known in the label noise literature, e.g., (Natarajan et al., 2013), (Ghosh et al., 2017)?
- Theorem 1: not clear from the theorem statement what the loss ell is. Certainly it will not hold without further assumptions on the loss (e.g., Lipschitz). I assume you must be referring to 0-1 loss.
- Theorem 1: please clarify explicitly how this compares to existing label noise generalization bounds, e.g., the Rademacher bound of (Natarajan et al., 2013).
- "Note the assumption holds generally and the bias-complexity trade-off does not exist if the assumption does not hold." -> this comment is fairly cryptic, and needs more explanation.
- "Corollary 1 implies that, for symmetric noise, a fixed encoder is likely better in high-noise settings." -> it is not a-priori clear how you are making the conclusion. You seem to be making assumptions on the VC dim of G o F versus G, and on the denominator on the RHS.
- "and converge to a poor local optimum" -> not clear. Corollary 1 says nothing about deep neural networks and local minima. It is not (obviously) the case that the global optimum under a fixed encoder would fare better.
- Theorem 2 -> please clarify what you are minimizing over.
- Theorem 2 -> please clarify if g, f in the LHS are chosen as the optimal solution.
- "the model will predict x to its clean label" -> this is highly surprising without further assumptions. Suppose the Bayes error is non-zero. Then the Bayes-optimal classifier itself will have non-zero population error. It does not seem possible for your method to avoid this, unless you assume separability of the data (i.e., zero Bayes error), which does not seem to be spelled out.
- Assumption 1 -> in Theorem 2, you assume an infinite N. So implicitly, are you assuming that the DNN model complexity scales with N, so that the model can interpolate the clean samples? Note that for a fixed model size, at some point one can increase the # of samples so that we are no longer in the overparameterized regime.
- Assumption 2 appears quite strong. I could only imagine a justification under the neural collapse phenomenon. As above, this would require model complexity to scale with N.
- Assumption 3 -> is this assumed for the optimal h, f; or every h, f; or something else?

**Note From Pc:**

if the above contains the word "oral" or "spotlight" please see: "oral" presentation means -> notable-top-5% and "spotlight" means -> notable-top-25%. As stated in our emails, we are disassociating presentation type from AC recommendations

**Summary Of Ac-Reviewer Meeting:**

See summary below. There was broad consensus that the paper had some impressive empirical results, although there were concerns on its completeness (missing baselines, and results on more challenging datasets). The following discussion with authors mostly clarified this, and scores were further raised. There was also the general sentiment that the theoretical results were not the main selling point of the paper.

_Attendees_

AC, Reviewers Y66C, DbNo, kNp2

Reviewers L66U and wiWY could not attend due to difficulties finding a suitable time-zone for all attendees.

_Meeting notes_

Reviewer Y66C: paper has interesting ideas. Main concern was around the experiments: inconsistent use of baseline method ELR from prior work. Response was confusing about why this is only reported in some settings. Also, results are limited to CIFAR10, CIFAR10/100-N and Clothing1M. Would like to see results on CIFAR100. Finally, some confusing points with the presentation of theoretical results. The latest author response includes some numbers with CIFAR100 and the ELR baseline, which seem impressive. However, the numbers appear to be quite a bit lower than what is usually reported in the literature.

Reviewer DbNo: agreed with the points of Y66C, particularly around the baseline and confusing presentation. The response does clarify somewhat. Willing to re-assess score.

AC: should we view the paper as a theoretical work with some experiments, or an empirical work with some theory? Reviewer Y66C and DbNo felt it is more the latter.

Reviewer kNp2: agreed that the theoretical results may not be strong enough to exactly explain what we observe in practice. However, saw them more as a qualitative guide to help ground the method, rather than a precise prescription. The basic idea seemed very interesting. The experimental results appear sufficient. While CIFAR100 would be nice, not clear that it is a hard requirement.

Reviewer Y66C: one point regarding CIFAR100 is that we may see quite different trends compared to CIFAR10. So, it would be good to assess. Authors have provided some initial results in the response, but would be great to have more.

AC: does CIFAR100-N suffice?

Reviewer Y66C: this tends to be much easier than the standard CIFAR100.